# Non-destructive transcriptomics *via* vesicular export

Niklas Armbrust [1,8], Martin Grosshauser [1,2,8], Julian Geilenkeuser [1], Luisa Stroppel[2], Mattea Jozinovic[2], Hella Levermann[2], Tobias Panne[2], Jannick Wißmann[2], Lukas Goelitz [2], Sebastian Schmidt[1,2], Tanja Orschmann[1,3], Ejona Rusha[3], Emily Steinmaßl[2], Florenc Widenmeyer [2], Niklas Warsing[2], Melike Sabry[2], Asina Sultanbai [2], Tobias Santl[1], Arie Geerlof [4], Oleksandr Berezin[2], Silviu-Vasile Bodea[1], Alessandra Moretti[5], Steffen Schneider [6], Fabian Theis [6], Julien Gagneur [7], Dong-Jiunn Jeffery Truong [1] ✉ & Gil Gregor Westmeyer [1,2] ✉

Transcriptomics enables comprehensive, multiplexed characterization of cellular states, yet prevailing methods typically require cell fixation or lysis, precluding longitudinal analysis of RNA expression in living cells. Here, we present non-destructive transcriptomics by vesicular export (NTVE), a platform for multi-time-point monitoring of RNA expression dynamics in living cells. Stabilized RNA reporter barcodes can be selectively packaged and exported from cells *via* virus-like particles (VLPs) bearing bioorthogonal affinity handles for convenient multichannel tracking of co-cultured cells. Using an engineered poly(A)-binding protein adapter, NTVE exports endogenous transcripts from inducible human and murine cell lines with high concordance to conventional lysate-derived RNA-seq. NTVE captures transcriptome changes in response to genetic and chemical perturbations within the same cells over time using standard sequencing workflows. NTVE can further be equipped with fusogens to deliver mRNA-encoded effectors or ribonucleoprotein gene editors from sender cells, activating gene reporters in co-cultured recipient cells. We demonstrate the utility of NTVE for monitoring hiPSC differentiation through daily non-destructive transcriptomic profiling of lineage-specific marker dynamics.

Transcriptomics, enabled by microarray and next-generation sequencing (NGS), has become the key technology for characterizing cellular states. Its unparalleled ability to amplify and multiplex biological samples has unlocked profound insights into cellular processes and molecular interactions. However, prevailing RNA sampling techniques are destructive and typically require cell fixation or lysis. Repeated sampling is currently only possible through laborious sequential aspiration using nanostraws or microneedles, which limits the number of robust cells and hinders scale-up[1,2].

[1]Institute for Synthetic Biomedicine, Helmholtz Munich, Neuherberg, Germany. [2]Department of Bioscience, TUM School of Natural Sciences, Technical University of Munich, Munich, Germany. [3]iPSC Core Facility, Helmholtz Munich, Neuherberg, Germany. [4]Institute of Structural Biology, Helmholtz Munich, Neuherberg, Germany. [5]TUM School of Medicine and Health, Technical University of Munich, Munich, Germany. [6]Institute of Computational Biology, Helmholtz Munich, Neuherberg, Germany. [7]TUM School of Computation, Information and Technology, Technical University of Munich, Munich, Germany. [8]These authors contributed equally: Niklas Armbrust, Martin Grosshauser. ✉e-mail: jeffery.truong@helmholtz-munich.de; gil.westmeyer@tum.de

Non-consumptive methodologies that enable multiple temporal analyses of transcriptomic changes within the same cellular population are therefore highly desirable, as they would permit longitudinal study of cells within their native network and environment.

Inspired by the information carried by circulating extracellular vesicles (EVs), researchers overexpressed secreted microRNAs (SecmiR) to generate artificial secreted biomarkers for cancer cells[3]. Similarly, the TRACE-seq method employed a protein handle fused to an exosome-resident CD9 to export m6A-modified RNAs via endogenous extracellular vesicles, enabling detection of gene expression changes upon hydrogen peroxide stress[4].

To engineer a bio-orthogonal export mechanism with adjustable strength, we previously constructed export vesicles derived from the HIV-1 gag polyprotein[5–7], to specifically export aptamer-tagged RNA barcode reporters via the PP7 coat protein (PCP) from cells[8]. Horns et al. developed a capsid-based export system named COURIER to secrete engineered barcodes from cultured cells for monitoring cell population dynamics[9]. While the fusion of a nanocage with MS2 coat protein (MCP) resulted in a substantial increase in the export of target RNA-specific barcodes, the authors noted that constructs based on MMLV-Gag resulted in the non-specific export of cellular transcripts. However, this transcriptomic information has not been used for longitudinal monitoring of alterations in cellular states[9].

In this study, we present an HIV-1-Gag-based system for exporting stabilized RNA barcodes and poly(A)-containing transcripts to enable multi-time-point transcriptomics from living cells (Fig. 1a). We have generated several inducible cell lines and an AAV system for non-destructive transcriptomics by vesicular export (NTVE) analyses of cocultures, non-invasive monitoring of transcriptomic responses to chemical perturbations, and non-invasive tracking of stem cell differentiation.

## Results

### HIV-Gag-based machinery enables RNA export from living cells

We utilized HIV-Gag as the main chassis to implement doxycycline (dox)-inducible membrane budding combined with specific adapters for transcript packaging and export (Fig. 1a). We inserted either a PP7 coat protein (PCP)[10] or the first two poly(A)-binding domains (RRM1 + RRM2)[11] of the PABPC1 protein downstream of the nucleocapsid (NC) domain of codon-optimized $Gag_{L21S}$ to direct the export towards PP7-tagged ($NTVE_{PCP}$) or polyadenylated RNA ($NTVE_{PABP}$), respectively. We designed an expression cassette for stable genomic integration of NTVE systems via an ecotropic lentivirus. A Pgk promoter-driven TetON3G transactivator enables dox-dependent TRE3G-driven expression of the export vehicle (Fig. 1b, Supplementary Fig. 1a, b). Furthermore, we introduced the mutation M161A to disrupt its binding to eIF4G and minimize interference with endogenous protein translation[12,13].

To achieve selective export of PP7 aptamer-tagged transcripts, we employed our previously established system[8] and targeted the endogenous housekeeping gene POLR2A through CRISPR/Cas9-mediated insertion of five PP7 loops into its 3′ untranslated region. Dox-induced expression of $NTVE_{PCP}$ yielded approximately 100-fold enrichment of POLR2A transcripts in the supernatant relative to a DMSO control (Supplementary Fig. 1c). Installation of a PP7 aptamer in its 3′ UTR resulted in a threefold increase in loading of POLR2A transcripts into NTVE particles compared with non-specific NTVE export (Supplementary Fig. 1d, e).

For intronic RNA export, transcript stability was enhanced through two complementary strategies (Supplementary Fig. 2a–c): enzymatic protection via catalytically inactive debranching enzyme ($DBR1_{H85A}$) and ribozyme-mediated circularization utilizing the Tornado system[14]. The truncated $miniDBR1_{H85A}$ variant competitively inhibits endogenous debranching activity, thereby preventing intron degradation and facilitating substantial PP7-tagged barcode export.

The Tornado-mediated circularization strategy demonstrated superior export efficiency, yielding higher barcode recovery than enzymatic stabilization alone (Supplementary Fig. 2d).

We next tested whether mRNA transcripts tagged with a PP7 aptamer could be exported to track the relative abundance of two cocultured cell populations via RNA reporters. To this end, cells were transfected with either mGreenLantern (mGL) or mScarlet-I (mSL) containing a PP7 aptamer in the 3′UTR. Twenty-four hours after transfection, the two cell populations were reseeded at defined ratios (10:1, 1:1, 1:10). Fluorescence was quantified by microscopy two days after NTVE induction. RNA was extracted from the supernatant for RT-qPCR quantification of the respective transcript abundances, which closely matched the stoichiometries obtained from microscopy (Supplementary Fig. 3a).

Next, we attached distinct affinity tags to the NTVE particles to separate distinct cell populations before transcript analysis. Based on the established role of the vesicular stomatitis virus glycoprotein (VSV-G) in mediating cellular uptake of virus-like particles (VLPs) via LDL receptors, we reasoned that it could serve as an effective membrane adapter for the incorporation of epitope tags. However, to avoid fusion between VSV-G and host cell membranes, we introduced the W72A mutation (dVSV-G) into the hydrophobic fusion loops[15].

We then fused HA ($NTVE^{HA}$) or FLAG epitope tags ($NTVE^{FLAG}$) to the N-terminus of dVSV-G and used magnetic anti-HA and anti-FLAG beads for pull-down. Matching abundances of barcoded transcripts were recovered from the co-culture population, confirming no carry-over (Supplementary Fig. 3b).

### $NTVE_{PABP}$ for export of endogenous transcripts

After inducing $NTVE_{PABP}$ budding with dox and enriching vesicles from the supernatant by ultracentrifugation, we observed a peak hydrodynamic radius of the vesicles of 65 nm (with the normalized distribution spanning ~30 to ~90 nm) by dynamic light scattering (DLS). In comparison, only very few scattering events were recorded from the ultracentrifuged supernatant taken from cells treated with DMSO vehicle control, yielding an apparent radius of <20 nm (normalized distributions in Fig. 1c). We then visualized the same samples by cryo-electron microscopy and observed membrane-enclosed vesicles with luminal contrast enhancement and a median radius of 67 nm ($n = 156$) (Fig. 1d, Supplementary Fig. 4). We then compared the exported transcripts from the $NTVE_{PABP}$ cell line after dox induction with those obtained by transient lipofection of the same genetic components using a standard RNA-seq analysis workflow (Supplementary Fig. 5a).

While transient overexpression resulted in substantial export of cytosolic transcripts, it also contained mitochondrial transcripts at comparable abundances, which should not be available for export via budding from the plasma membrane. In comparison, mitochondrial transcripts were strongly depleted in the supernatant from the $NTVE_{PABP}$ cell line (Fig. 1e). We therefore decided to work with cell lines harboring genomic integration of dox-inducible $NTVE_{PABP}$ to minimize potential membrane perturbations induced by lipofection.

For a more detailed characterization, we collected the supernatant from the inducible $NTVE_{PABP}$ cell line 72 h after dox induction and lysed the cells, subjecting both matched samples in triplicate to Illumina stranded mRNA library preparation and next-generation sequencing (NGS). We quantified the expression of genes detected with at least one count per million (CPM) from poly(A)-enriched samples and found high concordance between three $NTVE_{PABP}$ replicates and matching HEK293T lysates (Pearson $r = 0.95$, Spearman $\rho = 0.88$; p for both below numeric precision) (Fig. 1f).

In total, we detected 14522 genes in $NTVE_{PABP}$ that were also detected in the lysate. In comparison, only 1423 genes were detected exclusively in the lysate, leading to an intersection-over-union of 90.4% of all expressed genes (Supplementary Fig. 5b). As expected, genes

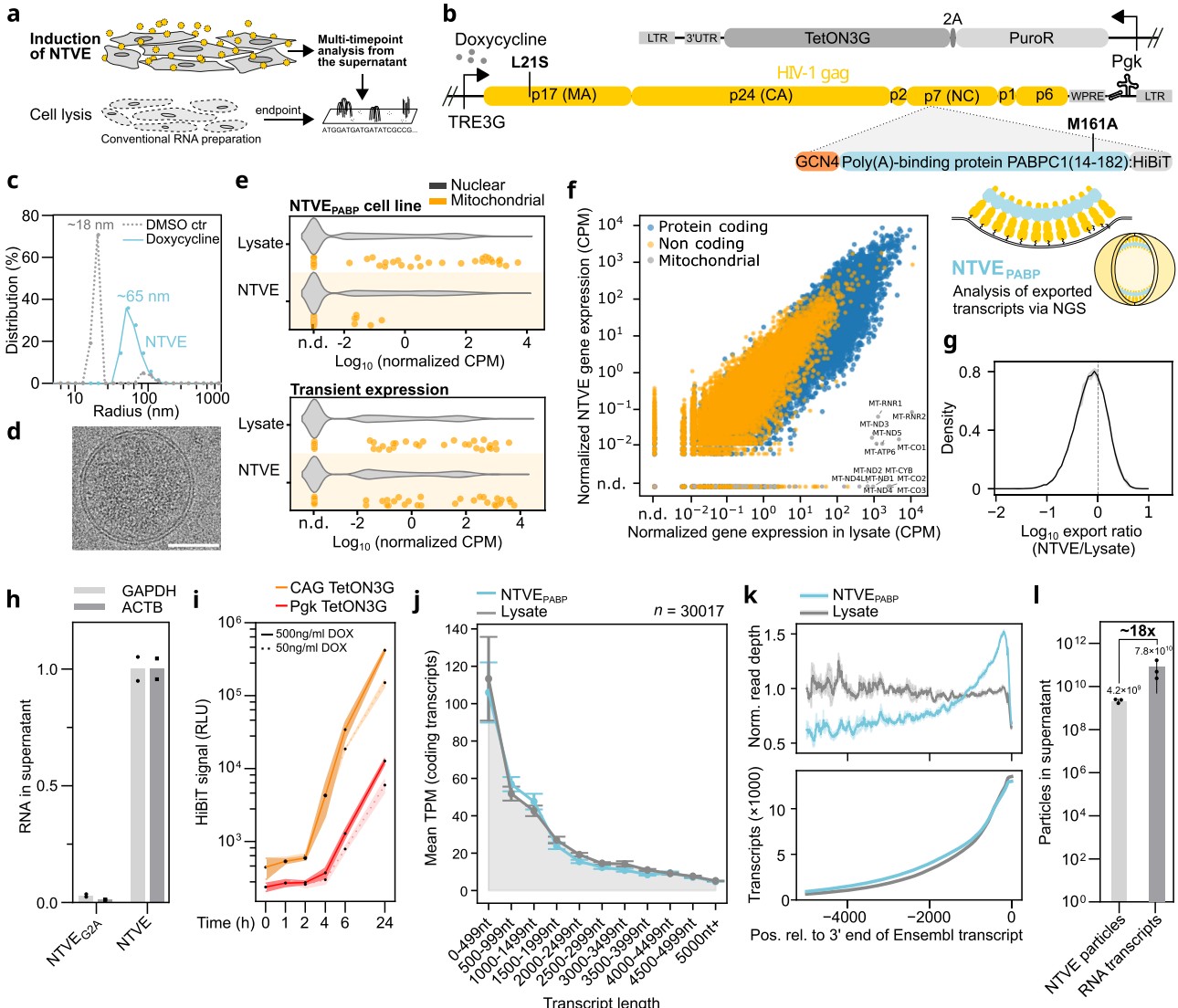

**Fig. 1 | Development of Non-destructive Transcriptomics by Vesicular Export (NTVE). a** Inducible expression of cellular RNA-export machinery enables repeated sampling of vesicle-enclosed transcripts from the supernatant of the same cell population. In contrast, conventional transcriptomics requires cell lysis at each time point. **b** Schematic of the transfer vector for generating stable NTVE cell lines *via* ecotropic lentiviral delivery of a modular cassette with a Pgk promoter driving puromycin *N*-acetyltransferase for selection (-1.1 kbp). A TetON3G transactivator (-0.7 kbp) is coupled *via* a 2 A peptide, enabling dox-dependent expression of non-infectious Gag mosaics that drive membrane budding. The NTVE cassette (-7 kbp) is flanked by two Long Terminal Repeats (LTR) and stabilized by the Woodchuck Posttranscriptional Response Element (WPRE), a triple helix, and a tRNA. The budding module (-2.1 kbp) can be equipped with different adapters for RNA packaging and affinity handles for purification. A HiBiT tag enables bioluminescence-based quantification of NTVE vesicles. The first two RNA recognition motifs (RRMs, -0.5 kbp) from poly(A)-binding protein PABPC1 enable the export of polyadenylated mRNAs, while the GCN4 leucine-zipper domain mediates dimerization. **c** Normalized distribution of hydrodynamic radii of NTVE vesicles after dox induction and enrichment by ultracentrifugation from the supernatant compared to a DMSO vehicle control (*n* = 1). **d** Cryo-electron micrograph of NTVE$_{PABP}$ particles enriched by ultracentrifugation. Scale bar: 50 nm (*n* = 1). **e** Distribution of expression levels for nuclear-encoded genes (grey violin plot) and mitochondrially encoded genes (orange dots) from a stable cell line with inducible expression of NTVE$_{PABP}$ (top) and from transient overexpression of the same construct (bottom). Elevated mitochondrial transcript levels in the supernatant indicate potential membrane perturbation. Reads were filtered for protein-coding genes (*n* = 3). **f** Normalized counts per million (CPM + $10^{-3}$ pseudocounts) from averaged NTVE vesicle triplicates collected in the supernatant of HEK293T cells 72 h

after dox induction, plotted against those from the corresponding lysates. Genes encoded on the mitochondrial genome are labeled with their gene symbol and plotted in grey (*n* = 3). **g** Distribution of log10 export ratios in HEK293T cells across detected protein-coding genes from three biological replicates processed identically (*n* = 3). The solid line shows the mean density across replicates, and the shaded band indicates the per-bin 95% confidence interval of that mean density (mean ± 1.96 × SEM across replicate histograms). **h** Quantification of representative mRNA transcripts in the supernatant for NTVE and the NTVE(G2A)$_{PABP}$ control, in which disrupted myristoylation prevents budding (*n* = 2, technical replicates). **i** Accumulation of NTVE particles in the supernatant after dox induction (continuous line: 500 ng/mL; dashed line: 50 ng/mL) monitored by bioluminescence from a HiBiT tag fused to the RRMs of PABPC1. Comparison of two stable cell lines driving the TetON3G transactivator *via* a Pgk or CAG promoter. Points indicate the mean of three biological replicates with *s.d.* bounds. **j** Mean counts per million (CPM) of expressed coding transcripts (CPM > 1) binned by transcript length, exported by NTVE$_{PABP}$, compared to the corresponding lysates. Input RNA was not enriched for poly(A); total RNA sequencing with rRNA depletion was applied. Data are plotted as the mean of triplicates ± *s.d.* **k** Average normalized read depth (top) and number of transcripts (bottom) as a function of nucleotide position relative to the 3' poly(A) site. Lines represent the mean ± *s.d.* of triplicates. **l** Quantification of the number of mRNAs per NTVE particle. NTVE$_{PABP}$ cells were incubated with 500 ng/mL doxycycline for 72 h. Bars represent the mean of biological replicates ± *s.d.*... mRNA was quantified by RNA-seq using spike-in of a known amount of in vitro transcribed mGreenLantern mRNA (*n* = 2); particle numbers were quantified by anti-p24 ELISA (*n* = 3). Input RNA for (**j**, **k**) was not enriched for poly(A); total RNA sequencing with rRNA depletion was applied. Source data are provided as a Source Data file. Analysis code is available *via* Zenodo deposition.

encoded by the mitochondrial DNA were depleted in the NTVE_PABP sample, including all 13 protein-coding mitochondrial genes (Fig. 1f and a full list in Supplementary Data 1).

The distribution of the export ratio (NTVE/Lysate) for all protein-coding genes is sharply peaked at parity, indicating that the majority of transcripts are represented at similar abundances in the NTVE supernatant and cell lysate, with limited selective enrichment or depletion (Fig. 1g, Supplementary Fig. 6).

To validate the dependence of transcript detection on dox induction, we measured the abundance of housekeeping gene mRNA in the NTVE fraction using RT-qPCR (Supplementary Fig. 7a) and compared the mRNA levels to those from a myristoylation-impaired Gag variant (G2A), which cannot bud off vesicles (Fig. 1h). Furthermore, we quantified NTVE_PABP particle export following dox induction *via* HiBiT bioluminescent signals, comparing moderate (Pgk promoter) and strong expression (CAG promoter) of the TetON3G transactivator. CAG-TetON3G produced a detectable bioluminescence signal ~4 h after induction, while Pgk-TetON3G required approximately six hours for detection (Fig. 1i). The quantified HiBiT luminescence can be used to estimate the amount of RNA exported into the supernatant (Supplementary Fig. 7b).

### Length-independent transcript capture *via* PABP adapter

We evaluated which transcript lengths were exported by NTVE_PABP vesicles, quantified transcript abundances using the total RNA seq workflow with rRNA depletion, and used Salmon[16] to calculate the average transcripts per million (TPM) for different length bins (Fig. 1j). The transcript length distribution in the NTVE fraction was highly similar to that observed in the lysate (Pearson $r = 0.993$, $p = 8.84 \times 10^{-10}$).

We then asked whether NTVE_PABP showed a read depth bias along the transcripts compared to the lysate samples. We found that the read depth of NTVE_PABP decreased as a function of the distance from the 3′ terminus, while that of the lysate remained constant over 5000 nt, consistent with the intended binding of the PABPC1-RRMs to the 3′ poly(A) tail of transcripts (Fig. 1k, top). The number of transcripts at each nucleotide position was not substantially different between conditions (Fig. 1k, bottom).

To characterize the effect of the PABPC1-RRMs on the profile of exported transcripts, we directly compared whole RNA preparations from the supernatant of the NTVE_PABP cell line with those from a line expressing Gag without an RRM (Gag) and a reference lysate subjected to nuclear depletion to enrich for cytosolic transcripts.

For NTVE_PABP, we again observed that read depth decreased with the position relative to the 3′ end of the Ensembl transcript annotation compared to the lysate. In contrast, Gag without RRMs showed a more uniform read depth across transcript lengths, suggesting less specific RNA binding compared to the PABP adapter (Supplementary Fig. 5c). When examining the length distribution of coding transcripts, NTVE_PABP reflected the lysate profile more faithfully than Gag-exported transcripts, which displayed a relative overrepresentation of lengths above 3000 nt (Supplementary Fig. 5d). This difference persisted when all 47,455 Ensembl-derived transcripts were included, yielding a high average TPM in the smallest length bin (Supplementary Fig. 5e). Moreover, we assessed the variance in NTVE transcripts and found a median coefficient of variation (CV) of 0.061, which was lower than that obtained from the corresponding lysate (CV = 0.133, Supplementary Fig. 5f, g).

We also investigated the effect of the commonly used inducer dox on differential expression in the reporter cells. Besides the expected overexpression of the Gag transcript, we observed upregulation of only a few metabolic genes known to be affected by dox[17] (Supplementary Fig. 8). As a final characterization, we determined the number of mRNAs per VLP, obtaining an average of ~18 mRNA molecules per particle (Fig. 1l, Supplementary Fig. 9).

### Optimized NTVE workflow

To improve sensitivity for low-input samples, we replaced the standard Illumina RNA-seq protocol with a custom, full-length cDNA amplification workflow (adapted from SmartSeq v2 and FlashSeq)[18,19], optimized for VLP vesicles rather than cells. In-house library preparation substantially increased throughput while reducing the required RNA input and maintaining reliable gene expression detection. Further efficiency was gained by combining lysis, reverse transcription, and pre-amplification into a single, one-pot reaction, which reduced hands-on time and reagent costs per sample. We also updated our analysis pipeline to accommodate the low-input library preparation. The pipeline implements a two-stage alignment and quantification: paired-end reads are aligned to a reference transcriptome *via* minimap2, followed by transcript quantification with Salmon. Read pairs are excluded if the proportion of unique k-mers in a read falls below 80% (Supplementary Fig. 10a, b). Quality control metrics paralleled those of the standard workflow, as reflected in the gene detection overlap between lysate and NTVE fractions (Supplementary Fig. 10c, Supplementary Data 1) and in the assessment of technical reproducibility (Supplementary Fig. 10d, e).

### Benchmarking NTVE

We next compared NTVE_PABP with TRACE-seq[4] and COURIER[9], all under identical transient Pgk-driven expression in HEK293T cells (Fig. 2, Supplementary Fig. 11). Quantification of total RNA from the respective supernatants demonstrated robust export from all Gag-based systems, whereas RNA export *via* endogenous EVs was lower by approximately two orders of magnitude (Fig. 2a). Assessment of mRNA detection rates (Fig. 2b) and the coefficient of variation (Supplementary Fig. 12a) across cumulative expression percentiles revealed that NTVE_PABP achieved superior transcript recovery across the abundance spectrum, maintaining the highest detection rate for highly expressed genes while also capturing low-abundance transcripts more effectively. We also analyzed the correlation between transcript abundance in the supernatant and lysate for binned transcript lengths and observed a higher correlation for NTVE_PABP for transcripts up to 1 kb (approaching 0.6) compared to the other methods (Fig. 2c).

Consistent with previous observations, the PABP adapter produced increased 3′ coverage of poly(A)-enriched transcripts in the NTVE_PABP fraction compared to export methods that do not use binders to the polyA tail (Supplementary Fig. 12b). We also found that removing large portions of the HIV-1 Gag MA/CA/NC regions (ΔMA12−114, ΔCA133−277, ΔNC), retaining only spacers that induce membrane curvature as well as the budding domain p6 (mini HIV-1 Gag_PABP)[10] resulted in a similar mRNA export profile. Increased 3′ coverage was likewise observed when PABP was exchanged with the engineered peptide sPAM2[20], which binds to the MLLE motif of endogenous PABPC1 (NTVE_sPAM), demonstrating its utility as a genetically compact alternative adapter (Supplementary Fig. 13).

### NTVE monitors genetic and chemical perturbations

Having established concordance between cellular and NTVE transcripts under normal growth conditions, we next evaluated whether NTVE could track changes in transcript expression in response to external stimuli. To evaluate the sensitivity and specificity of NTVE, we first induced *ASCL1*, a gene not expressed in HEK293T, *via* CRISPRa. After NTVE_PABP induction, cells were cultured for 72 h, and the supernatant was collected for NGS analysis of the exported transcripts. NTVE correctly detected *ASCL1* as the only significantly upregulated gene in the CRISPRa condition (Wald test, $p < 10^{-7}$; Supplementary Fig. 14).

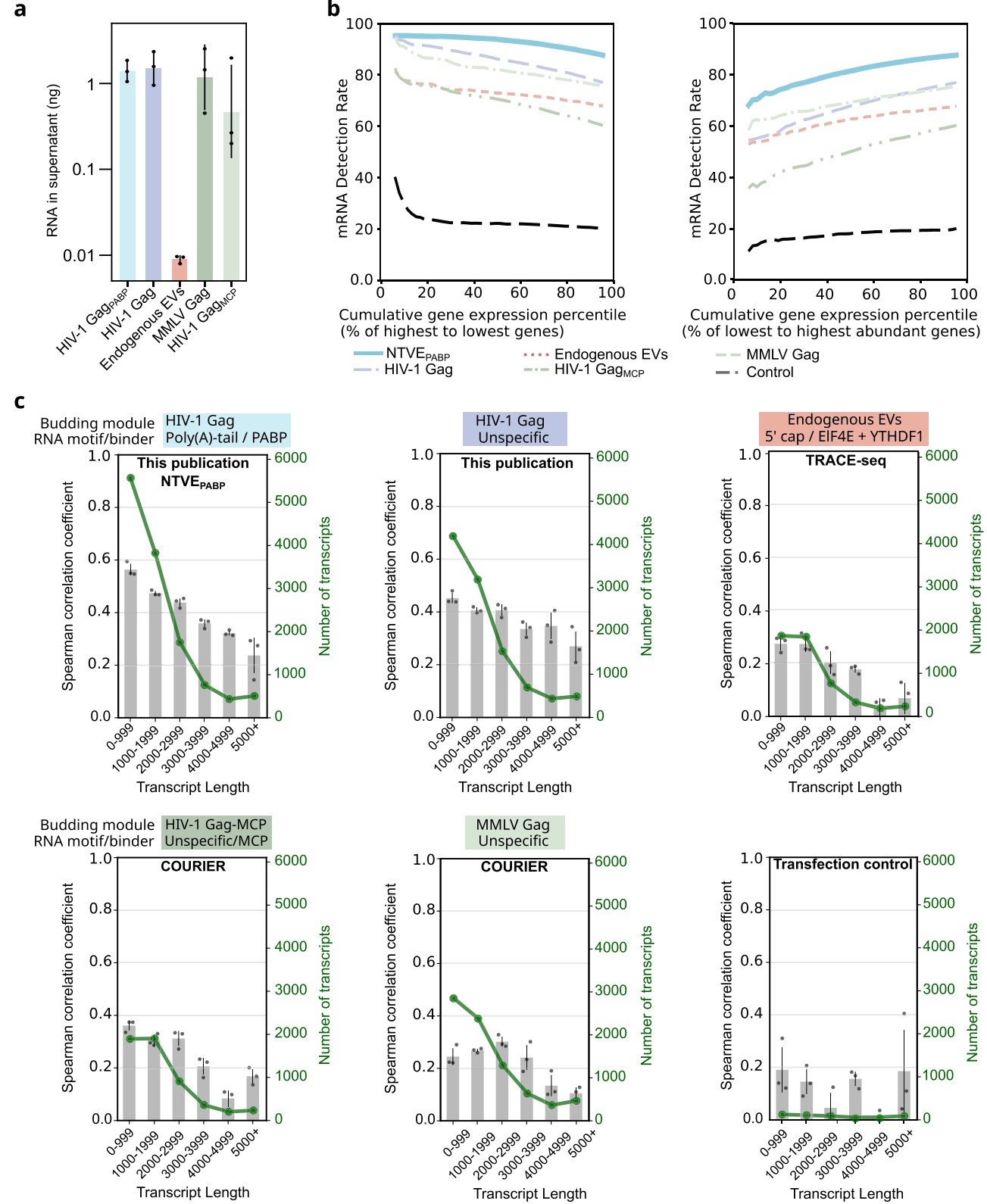

We then investigated stimulation with interferon-gamma (interferon-γ), which activates several essential signaling pathways in mammalian cells. We found high concordance (Pearson $r = 0.96$ in log-scale, $p < 10^{-127}$) between the fold changes of NTVE and lysate samples (Fig. 3a), and confirmed that mitochondrial transcripts were not enriched in the supernatant after perturbation (Supplementary Fig. 15). To assess how accurately NTVE classifies significantly up- or downregulated transcripts identified in the lysate, we constructed receiver operating characteristic (ROC) and precision-recall (PR) curves using $\log_2$ fold changes ($\log_2$FC) from NTVE as predictors for the binary classification of lysate-derived transcripts with absolute $\log_2$FC $> 1$ or $< 1$ and adjusted $p$-value $< 0.05$ (Wald test in DESeq2 package) (Fig. 3b, c). Pathway analysis of the differentially expressed genes confirmed that NTVE correctly identified the interferon-γ signaling network and co-activated pathways (Fig. 3d).

**Fig. 2 | Comparison of NTVE to other RNA export methods. a** Amount of RNA (mean ± s.d. of *n* = 3 biological replicates processed in triplicate) accumulated in the supernatant over 48 h following Pgk-driven expression of different export systems in HEK293T cells. **b** Transcript detection rate in all triplicates as a function of cumulative expression percentile for protein-coding genes (mitochondrial genes excluded), ranked from highest to lowest expression (left) or lowest to highest (right). **c** Export systems are named as in the referenced publications and categorized by budding module and RNA-binding module (MCP denotes non-specific RNA binding in the absence of MS2 aptamers). Mean Spearman correlation

coefficients of supernatant and lysate transcript abundances, binned by transcript length. Grey bars represent mean coefficients (*n* = 3 biological replicates; samples were processed in triplicate) with SD across matched replicate-pair correlations; the green line indicates the mean number of detected transcripts per bin. Genes with expression below 1 TPM were excluded. The large confidence intervals in the control transfection with NanoLuc luciferase (no VLP formation) reflect the low number of detected transcripts (<110 transcripts per bin). Raw sequencing data and analysis code are available *via* Zenodo deposition.

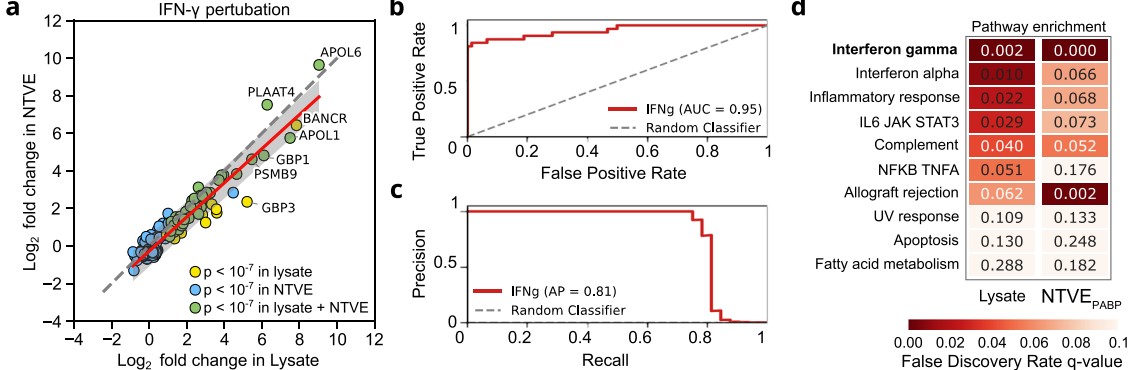

**Fig. 3 | Monitoring chemical perturbation with NTVE. a** Shrunken log2 fold changes (log2FC, apeglm) after IFN-γ perturbation obtained from NTVE plotted against lysate for genes significant in at least one compartment (*n* = 241 genes; supernatant collected 72 h post-perturbation). Colors indicate genes significant only in lysate (yellow, *n* = 15), only in NTVE (blue, *n* = 184), or in both (green, *n* = 42; DESeq2 adjusted *P* < 1 × 10$^{-7}$). Gene-level differential expression was tested separately for lysate and NTVE using DESeq2 negative-binomial Wald tests (*n* = 3 biological replicates per condition; two-sided); *P*-values were corrected for multiple comparisons by the Benjamini–Hochberg method, and apeglm shrinkage was applied to log2FC estimates. The red line shows the ordinary least-squares fit across all plotted genes; the grey band indicates the 95% prediction interval. The association between lysate and NTVE log2FC was assessed by a two-sided Pearson

correlation test (*r* = 0.955, *P* = 1.6 × 10$^{-128}$). The dotted line indicates perfect agreement (x = y) as a reference. **b** Receiver operating characteristic (ROC) and (**c**) precision-recall (PR) curves assessing the performance of NTVE log$_2$FCs as a classifier for identifying differentially expressed genes in response to IFN-γ stimulation. Genes identified in the lysate with an adjusted *p*-value < 0.05 (Wald test in DESeq2 package, Benjamini–Hochberg FDR correction) and absolute log$_2$FC >1 or <1 were designated as ground truth positives or negatives, respectively. NTVE-derived log$_2$FC values served as the predictor variable. The precision of the random classifier is 5‰. **d** Positively enriched pathways from the KEGG database after perturbation of HEK293T cells with 50 ng/mL IFN-γ. Pathways are ranked by ascending false discovery rate (FDR) *q*-values. Raw sequencing data and analysis code have been deposited on Zenodo.

## Inducible NTVE in primary mouse neurons

To demonstrate the utility of the dox-inducible NTVE system in primary cells, we co-transduced murine cortical neurons with a self-complementary AAV (scAAV) encoding the TetON3G transactivator and a corresponding AAV containing a TRE3G-driven NTVE$_{PABP}$ cassette, together with fluorescent proteins as a convenient readout of expression (Fig. 4a). NTVE expression was visible *via* fluorescence six days after transduction and two days post-treatment with 10 μM forskolin (Fig. 4b). Comparison of the transcriptome before *vs.* after treatment revealed strong upregulation of transcripts including *Bdnf*, *Crh*, *Scg2*, and *Sik1*, consistent with activation of the CREB pathway (Fig. 4c and Supplementary Data 2). Budding-deficient NTVE(G2A)$_{PABP}$ (Fig. 4d) and a DMSO vehicle control (Fig. 4e) yielded no detectable differentially expressed genes.

## Bio-orthogonal purification of NTVE vesicles

Next, we sought to apply NTVE to co-cultures of different cell types. We co-expressed the FLAG or HA surface handles described above (Supplementary Fig. 3b) on NTVE$_{PABP}$ vesicles and co-cultured human HEK293T cells and murine Neuro-2a cells expressing the respective surface tags (Fig. 5a). Enrichment of the respective NTVE species from the supernatant after dox induction with either anti-FLAG or anti-HA magnetic beads revealed a predominant representation of human or mouse transcriptomes, demonstrating selective transcriptome recovery from the target cell population (Fig. 5b). Correlation analysis of transcript profiles confirmed that FLAG-purified NTVE closely reflects both the

transcriptome of unpurified NTVE and the corresponding cell lysate (Supplementary Fig. 16).

## Vesicle-mediated genetic cell-to-cell communication

We next tested whether NTVE could be pseudotyped with glycoproteins for selective uptake by receiver cells expressing cognate receptors, thereby enabling engineered cell-cell communication. We co-expressed the MLV-Env glycoprotein (Eco) in NTVE$_{PCP}$-expressing HEK293T cells and co-cultured them with cells expressing a murinized SLC7A1, permissive for Eco[21]. We further engineered sender cells to package PP7-tagged mRNA coding for the large serine recombinase Pa01[22] and equipped the recipient cell with a matching attB/attP fluorescent switch that expresses mGL upon Pa01-mediated attB/attP recombination (Fig. 5c, Supplementary Fig. 17a). Flow cytometry analysis after dox induction revealed that a substantial number of receiver cells translated the Pa01 mRNA, mostly *via* NTVE uptake (bottom right quadrant) but occasionally also *via* direct fusion with a sender cell, indicated by simultaneous expression of the sender cell marker mTagBFP2 (upper right quadrant), also visible in confocal fluorescence microscopy (Fig. 5d, Supplementary Fig. 17b).

In addition, we tested whether NTVE could transmit information by packaging a CRISPR effector as a ribonucleoprotein (i.e., complexed with its guide RNA)[10]. We chose the improved prime editor (iPE-C)[23] and a pegRNA to target a reporter locus of recipient cells, resulting in a blue-to-green fluorescence switch (Fig. 5e, Supplementary Fig. 17c). For this system, we replaced the sender cell marker with the far-red fluorescent protein miRFP670nano3 and the MLV-Env glycoprotein

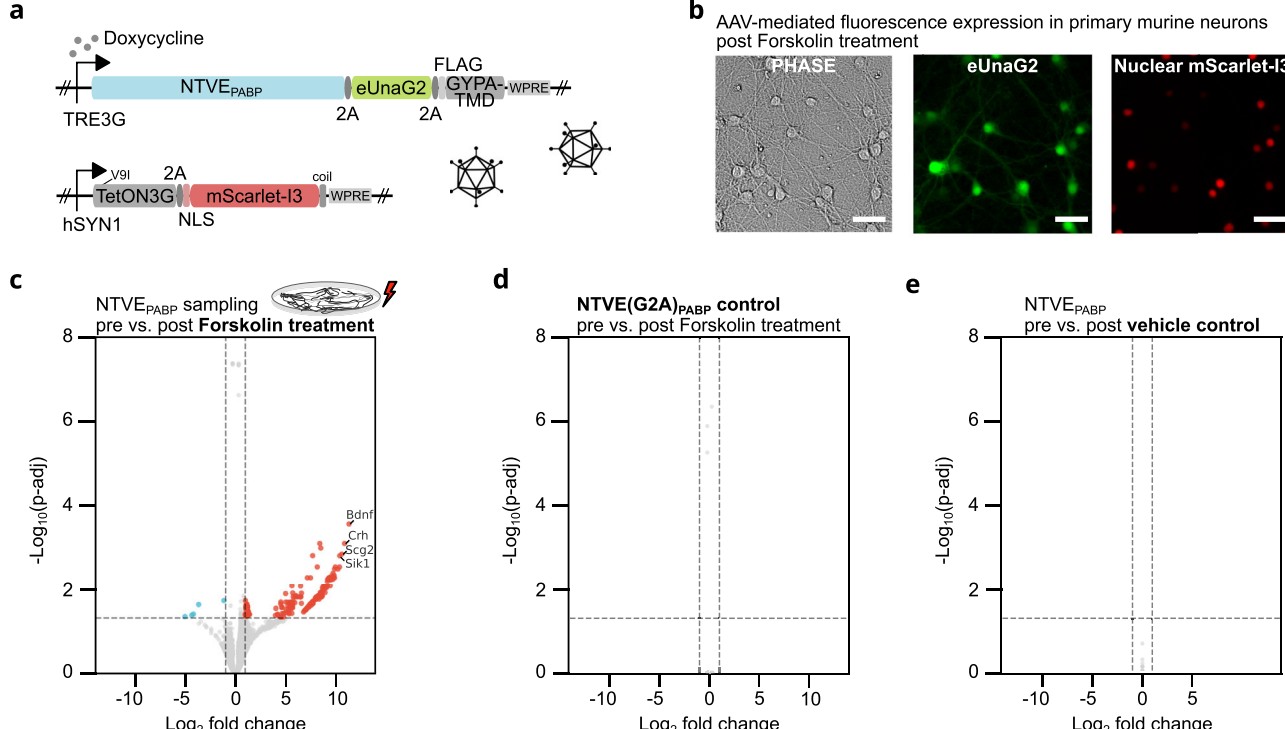

**Fig. 4 | NTVE monitoring in primary murine neurons. a** Schematic of the dual adeno-associated virus (AAV) system for dox-inducible NTVE expression in primary murine cortical neurons ($n = 3$). One vector delivers NTVE_PABP linked to the green fluorescent protein eUnaG2 and a surface-presented FLAG tag *via* P2A sequences. A second, self-complementary vector encodes the *hSYN1*-driven TetON3G transactivator and the red fluorescent protein mScarlet-I3, fused to a nuclear localization signal (NLS) *via* a P2A sequence. **b** Phase-contrast and fluorescence microscopy six days post-transduction and two days post-forskolin treatment. Scale bar: 200 μm.

**c** Differentially expressed genes ($n = 190$; Benjamini–Hochberg FDR correction; apeglm LFC shrinkage) detected by NTVE_PABP before and after treatment with 10 μM forskolin. RNA export was induced with 250 ng/mL doxycycline. **d** Control condition expressing the budding-deficient mutant NTVE(G2A)_PABP, in which disrupted myristoylation prevents vesicle formation. **e** DMSO vehicle control in which NTVE was induced, but neurons were not treated with forskolin. The horizontal dotted line represents $p = 0.05$. A complete list of differentially expressed genes is provided in Supplementary Data 2. Raw data and code are available *via* Zenodo.

---

with VSV-G to minimize direct membrane fusion between neighboring cells.

Remarkably, we observed a substantial number of editing events in the recipient cells by flow cytometry, suggesting that cell-cell communication can be engineered through the exchange of modular CRISPR effectors delivered as ribonucleoproteins, not just *via* mRNA.

### NTVE integration into hiPSC lines

Using the PiggyBac transposon system, we established NTVE_PABP-hiPSC lines by incorporating the doxycycline-inducible NTVE_PABP cassette into two well-characterized lines: SCTi003-A and WTC11 (UCSFi001-A) (Supplementary Fig. 18a, b). Analysis of hiPSC viability and cytotoxicity 48 hours after induction across a range of dox concentrations revealed no significant effect of NTVE_PABP expression (two-tailed t-test, $p = 0.217$ for the toxicity assay and $p = 0.089$ for the viability assay; Supplementary Fig. 18c).

We then differentiated the cells into the three germ layers to monitor lineage-specific markers[24]. Cells were cultured in lineage-specific differentiation medium supplemented with freshly prepared dox over 5–7 days (Fig. 6a and Supplementary Fig. 18d). HiBiT bioluminescence measurements confirmed sustained NTVE vesicle export throughout differentiation into all three germ layers (Fig. 6b and Supplementary Fig. 18e–j).

A UMAP embedding of samples with >100,000 aligned reads from three clones per lineage and all time points (Fig. 6c) showed dominant clustering by cell fate (germ layer identity), with individual clones displaying coherent temporal trajectories and low intra-lineage variability.

We monitored a standard set of surface markers by NTVE transcriptomics throughout the seven-day differentiation protocol. These markers are commonly recommended for combined surface protein labeling to identify each germ layer (StemMACS Trilineage Differentiation Kit). NTVE-derived transcript abundance profiles generally agree with endpoint flow cytometry analysis of the corresponding surface epitopes (Supplementary Fig. 19a–c). *SOX2* and *PAX6* transcript and protein levels were upregulated at the end of ectoderm differentiation, confirming expected lineage commitment. However, in the mesoderm condition, *CDH5* (CD144, VE-Cadherin) and *PDGFRβ* (CD140b) were detected in only ~30% of the population by flow cytometry in both the NTVE_PABP hiPSC line and the parental SCTi003-A line (Supplementary Fig. 19d–f). These low protein expression levels were consistent with the relatively low mesodermal transcript levels detected by NTVE (Supplementary Fig. 19b), which reported stronger expression in endoderm. Similarly, the recommended endoderm markers *CXCR4* (CD184) and *SOX17* were detected by flow cytometry but were also expressed at the transcript level in mesoderm (Supplementary Fig. 19c). The modest discriminatory ability of these standard markers prompted us to leverage the time-resolved transcriptomic profiles afforded by NTVE to identify improved lineage-specific candidates. We applied two complementary criteria: (i) differential expression across all time points between target and non-target lineages, and (ii) distinct temporal trajectories. This integrated analysis yielded a set of candidate markers with enhanced lineage specificity (Fig. 6d–f).

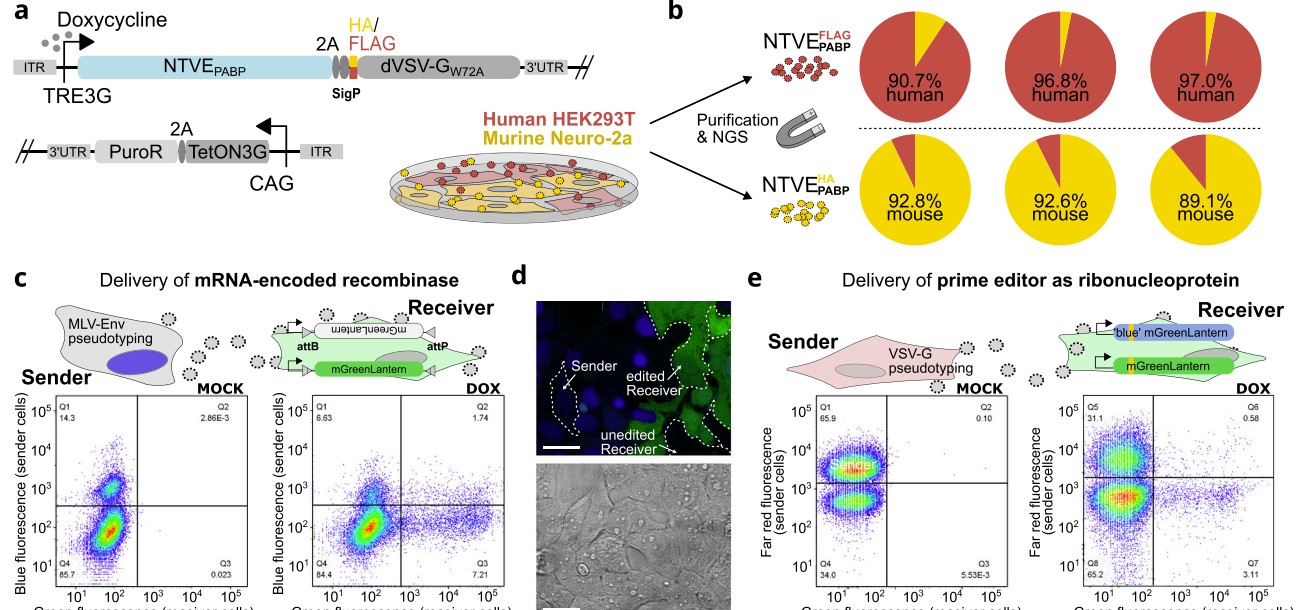

**Fig. 5 | Selective NTVE reporting and cell-to-cell communication in co-cultured cells. a** FLAG- or HA-tagged non-fusogenic dVSV-G(W72A) co-expressed with TRE3G-driven NTVE$_{PABP}$ serves as a surface handle for selective purification of NTVE vesicles from co-cultured murine Neuro-2a and human HEK293T cells. NTVE-mediated mRNA export was induced by the addition of 500 ng/mL dox. **b** NTVE vesicles were purified using anti-FLAG or anti-HA magnetic beads, followed by RNA isolation and NGS. Reads were mapped to human and mouse cDNA to quantify the proportion of species-specific transcripts. **c**–**e** Engineered cell-cell communication *via* vesicular export of either a recombinase-encoding mRNA (**c**, **d**) or a prime editor ribonucleoprotein (**e**) from sender cells into HEK293T receiver cells, inducing a genetic switch of a fluorescent reporter locus. **c** Vesicular export of PP7-tagged mRNA encoding the recombinase pa01$^{NLS}$ activates a mGreenLantern

reporter in receiver cells by site-specific inversion. Vesicles are pseudotyped with MLV-Env(ΔC16) glycoproteins for selective entry *via* a murinized SLC7A1 receptor on receiver cells. Flow cytometry scatter plots show the co-culture population of sender and receiver cells three days post (mock) induction of NTVE expression with dox. **d** Confocal microscopy of the co-culture from (**c**) after 72 h. Scale bar: 50 µm. **e** Delivery of a prime editor as a ribonucleoprotein (RNP) complex, causing a blue-to-green fluorescence shift upon editing of a reporter locus. Unlike (**c**), sender cells co-express the VSV-G glycoprotein to facilitate uptake by recipient cells *via* endogenous LDL receptors. Flow cytometry scatter plots show the co-culture population of sender and receiver cells three days post (mock) induction of NTVE expression with dox. Source data are provided as a Source Data file. Raw data and code are available *via* Zenodo.

## NTVE monitors cardiac differentiation

Next, we tracked cardiomyocyte differentiation by integrating NTVE$_{PABP}$ into a human induced pluripotent stem cell (hPSCreg MRIi003-A) line with demonstrated capacity for cardiac lineage commitment[25]. We monitored dox-induced NTVE export daily during both mesoderm and cardiac induction phases using HiBiT and collected NTVE vesicles each day (Fig. 7a). Visible contraction was first detected on day 6 post-induction and persisted until the end of the experiment on day 9 (Fig. 7b, Supplementary Movie 1).

To identify genes with significant temporal dynamics, we used DESeq2 to perform a likelihood ratio test comparing a natural cubic spline model against a time-invariant null model, yielding 146 transcripts with significant time-dependent expression (FDR < 0.05, Benjamini–Hochberg correction). Ranking these transcripts by time to peak expression revealed a cluster of cardiac marker genes whose upregulation coincided with the onset of visible contraction (Fig. 7c, Supplementary Fig. 20).

For temporally resolved differential expression analysis, we applied DESeq2 in a sliding window framework, comparing consecutive two-day bins (stride = 1). Per-gene fold changes from each comparison were ranked for gene set enrichment analysis (GSEA), revealing progressive enrichment of cardiac-specific pathways (Fig. 7c and list of top 10 hits in Supplementary Data 3). NTVE-derived expression profiles also recapitulated mesodermal-to-cardiac lineage trajectories (Supplementary Fig. 21a). Gene set enrichment analysis across the differentiation time course confirmed progressive activation of cardiac-specific pathways concurrent with the emergence of functional contraction (Supplementary Fig. 21b, c).

## Discussion

We expanded our established method of exporting intron-encoded RNA barcodes *via* RNA aptamers[8] to enable vesicular export of endogenous transcripts, thereby permitting non-destructive, multi-timepoint transcriptomic analyses in living cells.

Central to this approach is the capture of poly(A) tails by installing RRM domains from PABPC1 on the interior of the Gag protein shell. The resulting 3′-enriched read coverage confirms preferential binding to poly(A) tails (Fig. 1k, Supplementary Figs. 5c, and 12b). This binding mode likely accounts for the strong concordance between the length distribution of exported and intracellular transcripts. By contrast, Gag bearing its native nucleocapsid (NC) RNA-binding domain may preferentially export longer transcripts, consistent with a uniform binding probability across transcript bases that would statistically favor longer molecules carrying more potential binding sites (Fig. 2). As a genetically more compact alternative to the PABPC1 RRM domains, we also introduced the engineered peptide sPAM2[20], which non-covalently binds to endogenous PABPC1, resulting in similar efficacy (Supplementary Fig. 13).

Transient transfection is typically effective only in immortalized cell lines and carries the risk of increased membrane permeability and non-specific RNA leakage. (Fig. 1e). To circumvent these limitations, we generated stable mammalian cell lines, including two hiPSC lines, in which vesicular export is tightly regulated by dox induction. This strategy enables sampling of cellular transcripts without compromising membrane integrity by transfection and yields high concordance between NTVE-exported and lysate-derived transcriptomes, with strong depletion of mitochondrial

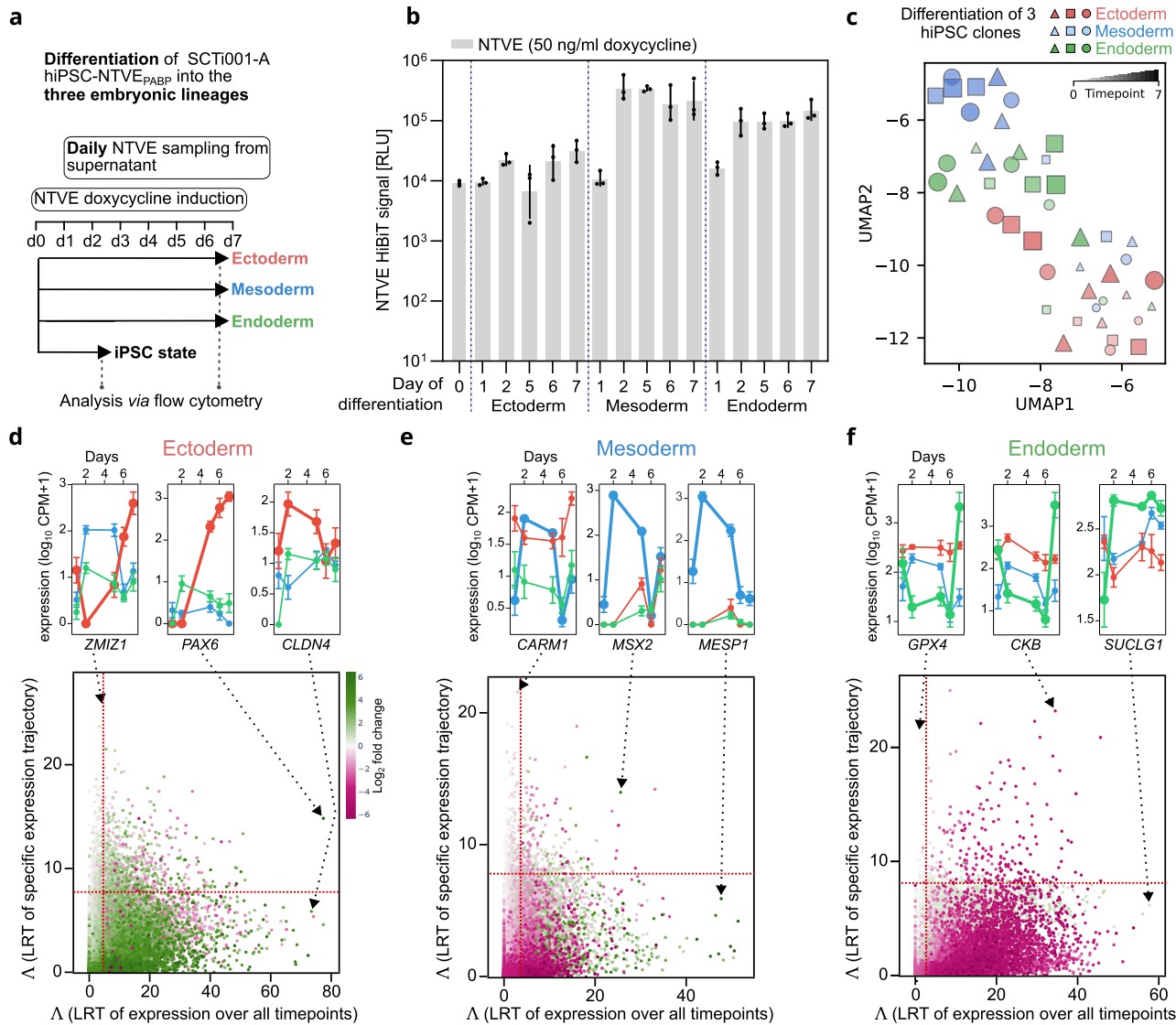

**Fig. 6 | Monitoring gene expression during trilineage differentiation.**
**a** Experimental timeline of trilineage differentiation of the SCTi003-A hiPSC line. NTVE-derived RNA was sampled daily from the supernatant, with NTVE budding continuously maintained by 50 ng/mL doxycycline (dox) throughout differentiation into ectoderm, mesoderm, and endoderm. Reference RNA was isolated from cell lysates at the hiPSC starting state and at the terminal time point for each germ layer. **b** HiBiT bioluminescence signal quantified from the supernatant throughout differentiation. Bars represent the mean of biological triplicates with *s.d.* **c** UMAP visualization of transcriptional trajectories during progression from the pluripotent state towards the three germ layers. Lineages are color-coded. Temporal progression is encoded by point size and transparency, with smaller, more transparent points corresponding to earlier differentiation time points. **d–f** Top: mean NTVE expression time courses ($n = 3$ biological replicates with error bars indicating the *s.d.*) for selected lineage-specific transcripts. Bottom: scatterplot of transcripts classified by two complementary likelihood ratio tests (LRT) identifying lineage-specific genes either *via* differential expression across all time points (x-axis, color-coded log-fold change) or *via* distinct temporal expression trajectories (y-axis). Dashed lines indicate the significance thresholds of the respective test statistic (Λ, LRT, $p < 0.05$). Raw sequencing data and code are provided *via* Zenodo deposition.

transcripts (Fig. 1e, f; Supplementary Figs. 11 and 15a). We also encoded NTVE$_{PABP}$ in an AAV vector to facilitate delivery to primary cells, including neurons (Fig. 4).

To monitor export efficiency, we incorporated a HiBiT bioluminescence reporter that permits estimation of optimal sampling intervals. A ~100-fold increase in HiBiT signal over baseline typically yields sufficient material for the detection of >10,000 genes (Supplementary Fig. 7b). Using spike-in normalization, we estimated an export rate of approximately 7000 transcripts per cell per day in HEK293T cells (Pgk-promoter driven TetON3G, lentiviral multicopy insertion). The hiPSC line (CAG-promoter driven TetON3G, PiggyBac-mediated multicopy insertion) exports at similar rates with ~8000 mRNA copies per cell per day.

By leveraging established template-switching-based RNA-seq protocols (FLASH-seq and SmartSeq), we achieved full-length amplification and higher sensitivity, detecting RNA from as little as 10 pg of input while maintaining a strong agreement between the NTVE fraction and the lysate.

We applied NTVE to monitor trilineage differentiation of hiPSCs and cardiomyocyte development, demonstrating its utility for tracking lineage-specific transcriptional dynamics over extended culture periods. NTVE can also be readily delivered to hard-to-transfect cells, such as primary murine neurons, *via* recombinant adeno-associated viruses (rAAVs).

Several avenues for further development merit consideration. Engineering larger vesicles may enable preferential sampling of longer

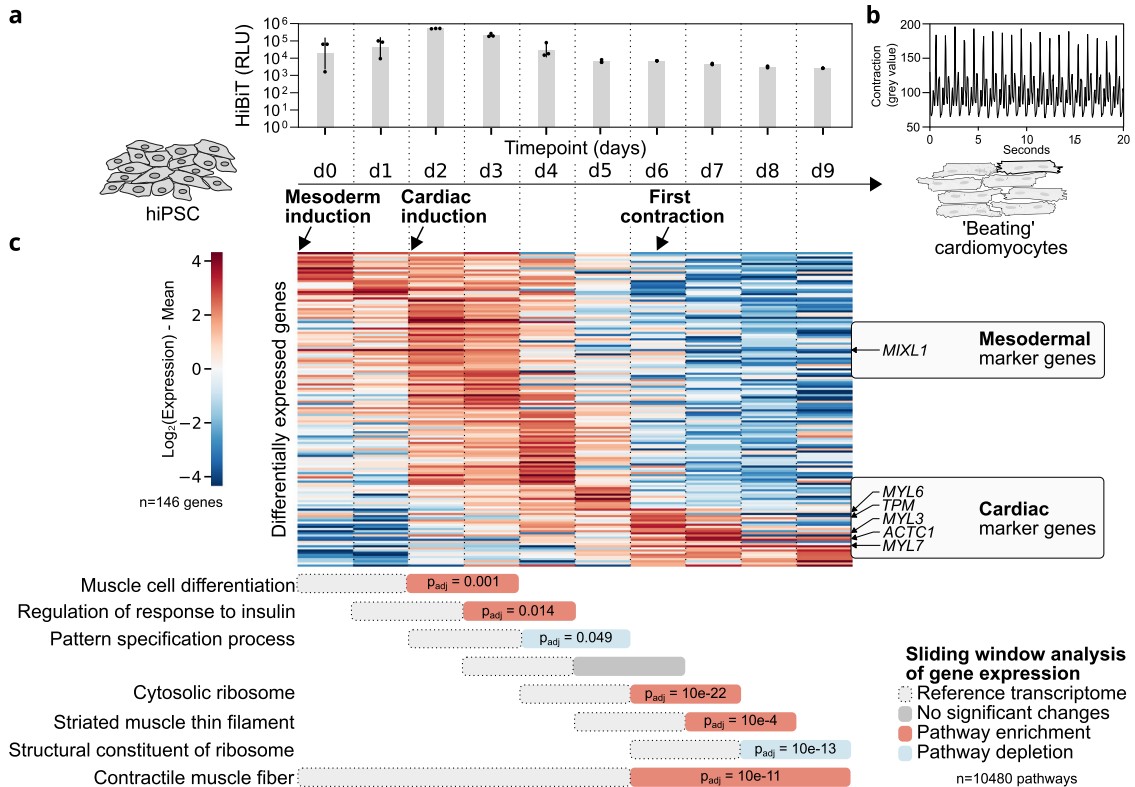

**Fig. 7 | NTVE captures transcriptional dynamics during hiPSC-to-cardiomyocyte differentiation.** **a** Directed differentiation of MRIi003-A hiPSCs into contracting cardiomyocytes. NTVE_PABP was continuously induced by the addition of 50 ng/mL dox. Export was quantified by measuring the HiBiT signal from the supernatant. The bars represent the mean of $n = 3$ biological replicates with error bars indicating the *s.d.* (top). Timeline of the cardiac differentiation protocol indicating media changes and the onset of visible contractions (bottom). **b** Quantification of cardiomyocyte contraction emerging on day 6 of the 9-day differentiation time course. The contraction frequency of ~1 Hz was derived from temporal signal fluctuations in the phase-contrast microscopy time series. **c** Heat map of row-centered log₂ expression values sorted by time to peak expression for all significantly differentially expressed genes during differentiation ($n = 146$; likelihood ratio test: natural cubic spline model versus a time-invariant reduced model; FDR < 0.05, Benjamini–Hochberg correction). For temporally resolved pathway analysis, a sliding window was applied in which each two-day bin was compared against the immediately preceding two-day bin, advancing with a stride of one day across the time course. Per-window fold changes were ranked for gene set enrichment analysis (GSEA); results were sorted by adjusted *p*-value, and the top-ranked pathway for each window is shown. A complete list of the top 10 hits for each comparison is given in Supplementary Data 3. Raw sequencing data and code are available *via* Zenodo deposition.

transcripts. Alternative RNA-binding domains beyond PABP and PCP could permit selective enrichment of specific transcript subsets. Additional membrane-exposed affinity handles, complementing the FLAG and HA tags employed here, would facilitate multiplexed analysis of diverse cell populations. Integration with microfluidic systems could enable continuous NTVE sampling during nutrient replenishment in three-dimensional cultures, supporting closed-loop optimization of differentiation protocols with real-time transcriptomic monitoring.

The repeated-measures design enabled by NTVE provides analytical strategies inaccessible to conventional endpoint sampling. Predictive models trained on paired NTVE–lysate correlations and sequential NTVE–NTVE trajectories could retrospectively reconstruct transcriptomic histories from terminal lysate data or prospectively forecast cellular states following perturbation. Similarly, comparison of NTVE transcripts before and after exposure to toxins or infectious agents could identify transcriptomic signatures associated with cellular resilience.

Beyond non-destructive transcriptomic readout, we demonstrated that mRNA and CRISPR effectors packaged as ribonucleoproteins can be exchanged between co-cultured cells. These capabilities suggest future applications in multi-timepoint proteomic and metabolomic profiling, as well as programmable intercellular biomolecule exchange.

In summary, NTVE constitutes an accessible and versatile platform for non-destructive, transcriptome-wide profiling of dynamic cellular processes. We demonstrate its utility in immortalized cell lines and human induced pluripotent stem cells, including daily monitoring throughout directed differentiation protocols. This validated performance in hiPSCs establishes the foundation for extending NTVE to organoid systems and ex vivo slice cultures, where non-destructive sampling and a repeated-measures design are particularly advantageous: each organoid can serve as its own longitudinal baseline control, directly addressing the substantial heterogeneity inherent to these models.

## Limitations of the study
NTVE cannot currently export nuclear-localized transcripts, including many non-coding and small nucleolar RNAs; however, RNA reporters exported *via* NTVE_PCP can provide information about the regulation of their respective promoters (Supplementary Figs. 1 and 2). Although we drove TetON3G expression from the CAG promoter − which provides strong activity in hiPSCs − sufficient NTVE induction should be confirmed by HiBiT luminescence assay for each differentiation protocol of interest (Supplementary Fig. 18e). Finally, while NTVE can export RNA barcodes reporting the relative abundance of co-cultured cell populations, it does not currently permit assignment of individual exported transcripts to single cells.

## Methods

### Ethical statement

All studies utilizing human induced pluripotent stem cells were conducted in accordance with applicable institutional and national regulatory guidelines.

### Molecular cloning

**Polymerase Chain Reaction (PCR).** Synthetic oligonucleotide primers (Integrated DNA Technologies) were prepared as 100 µM stocks in nuclease-free water. Genetic sequences were amplified from genomic or plasmid templates using Platinum SuperFi II PCR Master Mix (Thermo Fisher Scientific), following the manufacturer's recommended cycling parameters. Post-amplification, products were isolated *via* agarose gel electrophoresis and recovered using the Monarch DNA Gel Extraction Kit (New England Biolabs; NEB).

**Restriction digestion with endonucleases.** Standard DNA digestions were performed using NEB restriction endonucleases in 40 µL reaction volumes containing 1–3 µg of plasmid DNA, strictly adhering to the supplier's enzymatic protocols. Digested fragments were resolved on agarose gels and purified using the Monarch DNA Gel Extraction Kit (NEB).

**Gibson assembly and ligation.** For assembly, DNA concentrations were quantified *via* spectrophotometry (NanoDrop 1000, Thermo Fisher Scientific). Ligation reactions (10 µL) utilized 30–50 ng of the origin-containing backbone and a 1:1–3 molar ratio of backbone to insert, facilitated by the Quick Ligation Kit (NEB) for 5–10 min at room temperature. Gibson assemblies were executed in 15 µL volumes using 50 ng of backbone DNA and 1:1–5 molar ratios of backbone to insert. These reactions employed the NEBuilder HiFi DNA Assembly Master Mix (NEB) with incubation at 50 °C for 20–60 min.

**DNA agarose gel-electrophoresis.** DNA fragments were analyzed on 1% (w/w) agarose gels (Standard Agarose, Carl Roth) cast in 1x TAE buffer and supplemented with 1:10,000 SYBR Safe (Thermo Fisher Scientific). Electrophoretic separation was conducted at 100 V for 20–40 min. Size verification was performed relative to the 1 kb Plus DNA Ladder (NEB), and samples were prepared using 6x Purple Gel Loading Dye (NEB).

**Bacterial strains (E. coli) for molecular cloning.** *E. coli* NEB Stable chemically competent cells were utilized for all cloning and transformation. Successful transformants were selected using 100 µg/mL carbenicillin (Carl Roth). Routine bacterial growth was carried out in Lysogeny Broth (LB) or on LB agar plates, each supplemented with appropriate antibiotics.

**Bacterial transformation with plasmid DNA.** Competent cells (50 µL) were thawed on ice and combined with 1–5 µL of ligation or Gibson assembly mixtures. Following a 30 min ice incubation, cells were heat-shocked at 42 °C for 30 seconds, returned to ice for 5 min, and recovered in 450 µL SOC medium (NEB). Suspensions were then spread on selective agar plates and incubated overnight at 37 °C.

**Plasmid DNA purification and Sanger sequencing.** Candidate colonies were picked into 2 mL of selective LB medium and cultured at 37 °C for a minimum of 6 h. Plasmid DNA was isolated using the Monarch Plasmid Miniprep Kit (NEB) and verified through Sanger sequencing (GENEWIZ, Azenta Life Sciences). Sequence-verified clones were subsequently expanded in 100 mL selective LB medium overnight, and high-yield plasmid DNA was harvested using the QIAGEN Plasmid Maxi Kit.

### Cultivation of mammalian cell lines

Cells were maintained at 37 °C in a humidified atmosphere with 5% $CO_2$.

HEK293T (ECACC 12022001) and Neuro-2a (ECACC 89121404) lines were cultured in Advanced DMEM (Gibco) supplemented with 10% FBS, GlutaMAX, 100 µg/mL penicillin-streptomycin, 10 µg/mL piperacillin, and 10 µg/mL ciprofloxacin. Upon reaching 90% confluence, cells were washed with DPBS and dissociated using Accutase (Gibco) for 5–10 minutes. Enzymatic activity was neutralized with pre-warmed complete medium before cells were quantified and re-seeded into various formats (6-, 48-, or 96-well) for downstream induction or transfection.

**Human induced pluripotent stem cells.** Human induced pluripotent stem cells (hiPSCs; WTC11, SCTi003-A, MRIi003-A) were handled in accordance with institutional and national ethics guidelines. Cells were grown on Geltrex-coated surfaces in Essential 8 (E8) Flex medium. At ~70% confluence, hiPSCs were passaged using StemMACS Passaging Solution XF (Miltenyi Biotec) or Accutase (Gibco) at room temperature for 5 min, followed by manual dissociation in E8 Flex medium.

**Primary murine cortical neurons.** Murine primary neurons (Gibco Thermo Fisher Scientific, A15586) were recovered according to the manufacturer's protocol and maintained on PDL-coated (100 µg/mL) plates. The neurons were cultured in Neurobasal media (Thermo Fisher Scientific) supplemented with B27plus (Thermo Fisher Scientific), CultureOne (Thermo Fisher Scientific), GlutaMAX (Thermo Fisher Scientific), and Antibiotic-Antimycotic (Thermo Fisher Scientific). Twice a week, half the volume was replaced with fresh media for maintenance.

For AAV transduction, cells were seeded into a 12-well plate at a density of 280,000 cells per well. After 3 days in culture, cells were transduced with a total of 10 µL of crude AAV concentrate of each AAV production (scAAV hSyn-TetOn(V9I): $3 \times 10^{12}$ vg/mL; AAV Tre3G-NTVE: $1 \times 10^{11}$ vg/mL; AAV Tre3G-NTVE (G2A)$_{PABP}$: $7 \times 10^{12}$ vg/mL). Transgene expression was induced 3 days after transduction with 250 ng/mL doxycycline. Successful delivery was verified by fluorescence microscopy for both reporter genes mScarlet and eUnaG. Stimulation with 10 µM forskolin was carried out 2 days after doxycycline addition.

**Genetic constructs and generation of stable NTVE cell lines.** The NTVE knock-in plasmid is based on random integration *via* ecotropic lentiviral delivery or PiggyBac transposase activity. For transposition, an NTVE donor plasmid was co-transfected with a plasmid encoding a hyperactive mutant of the PiggyBac transposase[26,27]. For ecotropic pseudotyping of the lentiviral delivery, the ecotropic MLV glycoprotein (Addgene #35617) was used in combination with psPAX2 (Addgene #12260) and a transfer vector encoding NTVE. In both cases, the puromycin selection cassette is used to select for multi-copy insertion (4 µg/mL for HEK293T and Neuro-2a; 1 µg/mL for hiPSC). All core plasmids will be available from Addgene.

**Ecotropic lentiviral delivery (HEK293T).** Cells were seeded on a 6-well plate in 3 mL advanced DMEM with a concentration of 250,000 cells/mL. The cells were transfected one day post-seeding with a total of 1.2 µg plasmid DNA with the following stoichiometry: psPAX2 packaging plasmid (Addgene #12260, 400 ng), pseudotyping plasmid pCMV-Eco (adapted from Addgene #12260; 200 ng), and the transfer plasmids encoding psi-tagged, LTR-flanked NTVE (600 ng). Lipofection was performed according to the manufacturer's protocol (jetOPTIMUS, Polyplus). 72 h post-transfection, the supernatant was collected, sterile filtered (0.45 µm), and concentrated (~20x) using a 100 kDa cut-off filter. For transduction, the HEK293T WT cells were seeded on a 48-well plate (75,000 cells in 500 µL advanced DMEM). One day post-seeding, cells were treated with the ectotropic booster

according to protocol (Takara Bio, 631471) before the supernatant of one 6-well production was applied to three wells. Cells were expanded to a T25 flask three days post ecotropic transduction and selected for multi-copy insertion with 4 µg/mL puromycin.

For the ecotropic delivery, the TetON3G inductor was driven by a Pgk promoter. The construct was used in the following figures: Fig. 1c–k and Supplementary Figs. 1, 7a, 8b, 14.

**Random integration *via* PiggyBac transposase (HEK293T and Neuro-2a).** Cells were co-transfected (jetOPTIMUS, Polyplus) with plasmid DNA in the following stoichiometry: 1 part of the plasmid encoding a hyperactive mutant of the PiggyBac transposase[26,27] and 2 parts of the NTVE knock-in plasmid. Cells were expanded three days post transfection or ecotropic transduction and selected for multi-copy insertion with 4 µg/mL puromycin. The construct was used in the following figures: Figs. 1i, l, 3, 4a, and Supplementary Figs. 3, 4, 5, 9, 10, 15, 16.

**Generation of a monoclonal, stable hiPSC NTVE_PABP cell line (WTC11, SCTi003-A, MRIi003-A).** The hiPSC NTVE_PABP cell lines were generated by multi-copy random integration *via* PiggyBac transposase. For seeding, hiPSCs were singularized using Accutase (Sigma) for 5 min at 37 °C and counted. The cells were seeded at a defined cell number in E8 flex medium supplemented with 10 µM Y27632 (Enzo Life Sciences) on a Geltrex-coated 48-well plate (40,000 cells in 250 µL). 24 hours post-seeding, cells were supplemented with 300 µL fresh media without Y27632 and subsequently transfected according to protocol with 750 ng plasmid DNA using the Lipofectamine Stem Reagent (Thermo Fisher Scientific). Cells were allowed to recover to ~70% confluency in E8 flex medium. NTVE hiPSCs were clonalized and picked; green fluorescent colonies were expanded.

**Generation of AAVs for inducible expression of NTVE.** (relates to Fig. 4)

HEK293-T cells were seeded in 100 mm dishes and transfected with 3 µg each of three plasmids for AAV production: the helper plasmid pAdDeltaF6 (addgene #112867), a RepCap plasmid (pAAV2/1, addgene#112862), and the cargo plasmid encoding the Tre3G-NTVE_PABP with eUnaG as green reporter or hSyn-TetOn(V9I) with mScarlet-I3 as red reporter. The transfection mix also included 0.5 µg of each of two microRNAs targeting HIV-1 Gag and WPRE (5% of total DNA) to reduce leakage NTVE expression in the producer cells. As a negative control, the budding-deficient NTVE(G2A)_PABP variant was used.

HEK293T cells were seeded in advanced DMEM containing 2% FCS. 72 h post-transfection, the culture medium and the production cells were harvested. After vortexing thoroughly, supernatants were cleared and sterile filtered (0.22 µm PVDF sterile filter, Sigma-Aldrich). Subsequently, the supernatants were concentrated using 100 kDa MWCO Amicon centrifugal concentrators (Merck) to about 100 µL by spinning at 3000 g for 30 min. Concentrated supernatants were stored at 4 °C until further use.

**Generation of stable HEK293T receiver cell lines for delivery of an mRNA-encoded recombinase (relates to Supplementary Fig. 17).** The polyclonal recombinase reporter cell line was designed by integrating an inverted mGL codon sequence flanked by inverted attB/P sites downstream of a CAG promoter onto a PiggyBac transposon. A puromycin resistance marker was also placed under the control of the same promoter *via* an IRES downstream of the reporter. HEK293T cells were co-transfected with a plasmid containing the entire transposon and a corresponding PiggyBac transposase to stably integrate the transposon into the HEK293T genome (600,000 cells, 1.2 µg plasmid DNA, jetOPTIMUS, polyplus), and selected with 1 µg/mL puromycin for at least 14 days.

**Generation of stable HEK293T receiver cell lines for delivery of a prime editor RNP (relates to Supplementary Fig. 17).** To monitor prime editing events in the receiver cell, a reporter encoding the blue-shifted mGL mutant (G65S, Y66H) with a CAG promoter was cloned into flanking homology arms for a safe-harbor locus. To generate the stable HEK293T reporter cell line, cells were transfected with the donor plasmid as well as a second plasmid encoding Cas9 and the corresponding gRNA according to the manufacturer's protocol (600,000 cells, 1.2 µg DNA, jetOPTIMUS, Polyplus) 24 hours post-seeding on a 6-well plate (600,000 cells in 3 mL per well) in the presence of 0.5 µM of the DNA-PKcs inhibitor AZD7648 (HY-111783; MedChemExpress) and CAG-promoter-driven i53, a 53BP1 inhibitor. The polyclonal population was monoclonalized using limited dilution in 96-well plates, and clones were genotyped to determine homozygosity. To generate susceptibility for NTVE particles pseudotyped with MLV-Env, the cell line was further modified in the endogenous SLC7A1 locus *via* prime editing to express an engineered variant of the SLC7A1 protein with a murinized binding site (238-244(KEGKPGV > NVKYGE)). The resulting polyclonal cell line was monoclonalized by limited dilution, and clones were tested for ecotropic mGL delivery *via* integrase-deficient lentivirus (IDLV). Green fluorescent clones were expanded. The pegRNA sequence can be found in Supplementary Data 4.

**Generation of stable HEK293T sender cell lines for delivery of mRNA or prime editor RNP (relates to Supplementary Fig. 17).** The polyclonal sender cell lines were designed by combining dox-inducible expression of NTVE components (gag-PCP, glycoprotein) with constitutive expression of TetON3G transactivator, a selection and fluorescence marker of choice, and either the PP7-tagged mRNA or gene-editor-RNP cargo onto a PiggyBac transposon (see Supplementary Fig. 17 for a detailed schematic). Sender cell lines were then generated by transfecting and selecting HEK293T cells with 1 µg/mL puromycin.

**Co-cultivation of sender and receiver cells (relates to Supplementary Fig. 17).** Sender and receiver cells were seeded in a 1:1 ratio onto poly-D-lysine-coated 2-well slides (Ibidi) in 1.2 mL medium at a combined density of 300,000 cells/mL and then co-cultivated for 72 hours either with or without dox before being analyzed with confocal microscopy and flow cytometry.

**ASCL1 transactivation (relates to Supplementary Fig. 14).** For stable upregulation of *ASCL1*, 600,000 HEK293T:NTVE_PABP cells were seeded on a 6-well plate and transfected with 1.2 µg plasmid-DNA encoding an ITR-flanked, EF1a-driven dCas9(D10A;H840A):VPR[28] and four U6-driven sgRNAs targeting the transcription start site[29] (800 ng) and the hyperactive PiggyBac transposase (400 ng).

−886 ntGTTGTGAGCCGTCCTGTAGG
−572 ntGCCAATTTCTAGGGTCACCG
−451 ntGAGAACTTGAAGCAAAGCGC
−196 ntGGGGAGAAAGGAACGGGAGG

**IFN-γ perturbation (relates to Supplementary Fig. 15).** HEK293T:NTVE_PABP cells were seeded on pre-coated (poly-D-lysine) 6-well plates at a density of 750,000 cells in 3 mL advanced DMEM supplemented with 50 ng/mL interferon gamma. One day post-seeding, the NTVE system was induced with 500 ng/mL dox and cultured for three days before RNA was prepped from the supernatant and cell lysates.

**Trilineage differentiation of hiPSC (relates to Fig. 6 and Supplementary Fig. 18).** Differentiation of WTC11:hiPSC:NTVE_PABP into ecto-, meso-, and endoderm was performed on 48-well plates (500 µL/well feed volume) using the STEMdiff Trilineage Differentiation Kit (StemCell Technologies) according to the

manufacturer's instructions. For ectoderm and endoderm differentiation, 200,000 cells were seeded, while 50,000 cells were seeded for differentiation into mesoderm. Cells were seeded as single cells in E8 Flex media supplemented with 10 μM Y27632. One day post-seeding, cells were washed using DPBS, and subsequently, pre-warmed differentiation media were added to the cells. This procedure was repeated each day; the supernatants were collected, and the RNA from the NTVE fraction was purified for analysis via RT-qPCR. The differentiation media were supplemented with 50 ng/mL dox throughout the entire course of differentiation.

Tri-lineage differentiation of SCTi003-A:hiPSC:NTVE$_{PABP}$ was performed on 24-well plates using the StemMACS™ Trilineage Differentiation Kit for human cells (Miltenyi Biotech) according to the manufacturer's protocol. The cells were seeded in E8/Flex cultivation medium with ROCK inhibitor using cell densities of 80,000 cells/well for ectodermal and endodermal, and 60,000 cells/well for mesodermal differentiation. After one day, NTVE export was induced with 50 ng/μL dox. One day post-dox-induction for mesodermal or two days for endo- and ectodermal differentiation, the medium was changed daily according to protocol, and supernatants were collected for library preparation.

As a control condition, cells were cultured in E8 flex media to maintain iPSC status. Cells were sacrificed for RNA isolation at the end of the differentiation. All supernatants were analyzed for NTVE export strength via HiBiT luminescence assay. After differentiation, cells were stained to analyze the expression of lineage-specific markers by flow cytometry. Cells were gated according to forward scatter area (FSC-A) versus side scatter area (SSC-A) (P1) and forward scatter height versus forward scatter area (FSC-H vs FSC-A; P2). Green- and red-fluorescent populations were defined by quantifying fluorescence (P3; see exemplary gating strategy in Supplementary Fig. 22).

The following antibodies were used: PAX6 (APC, REAfinity™), SOX2 (FITC, REAfinity™), SOX17 (APC, REAfinity™), CD184/CXCR4 (Vio® Bright FITC, REAfinity™), CD144/VE-cadherin (FITC, REAfinity™), and CD140b (APC, REAfinity™) (all anti-human; Miltenyi Biotec).

**MRIi003-A differentiation into contracting cardiomyocytes (relates to Fig. 7 and Supplementary Figs. 20 and 21).** Cardiomyocyte differentiation of MRIi003-A:NTVE$_{PABP}$ hiPSCs was performed on 12-well plates using the StemMACS™ CardioDiff Kit XF for human cells (Miltenyi Biotech, 130-125-289) according to the manufacturer's protocol. 800,000 cells/well were seeded in Mesoderm Induction medium (day 0). Cells were cultured in Cardiac Cultivation medium, and cardiac formation was induced on day 2 with Cardiac Induction medium. NTVE was induced each day by supplementing the medium with 50 ng/μL dox. Medium was changed daily, and supernatants were sampled for library preparation. NTVE export strength in supernatants was analyzed via HiBiT luminescence assay.

**Viability and toxicity assays (relates to Supplementary Fig. 18c).** To assess viability and cytotoxicity, NTVE hiPSC were seeded as single cells (25,000 cells) on a Geltrex-coated 96-well plate in 100 μL E8 flex media supplemented with 10 μM Y27632 (Enzo Life Sciences, ALX-270333-M005). One day post-seeding, cells were washed with DPBS, and the NTVE system was induced with different dox concentrations.

The CellTiter-Glo 2.0 Cell Viability Assay (Promega, G9241), as well as LDH-Glo Cytotoxicity Assay (Promega, J2380), was performed in triplicate according to the manufacturer's protocol 48 hours post-induction.

**Characterization of NTVE particles**
**Dynamic light scattering (DLS) analysis (relates to Fig. 1c).** Dynamic Light Scattering (DLS) measurements were performed using a DynaPro NanoStar DLS system (Wyatt Technology). The supernatant from NTVE$_{PABP}$ HEK293T cells was collected 48 hours post-dox induction or addition of DMSO vehicle control. Samples were sterile filtered (0.45 μm) to remove any large particulates that could interfere with the measurement and concentrated via ultracentrifugation (120,000 × g, 2 h). Pellets were resuspended in PBS (1/100 of the original volume). For each measurement, 2 μL of the sample was loaded into a clean, dust-free quartz cuvette (WTC JC-325). The cuvette was carefully inspected for any scratches or imperfections before use. The sample was allowed to equilibrate to 25 °C prior to data acquisition. Between measurements, pure PBS samples were run to verify that there was no remaining signal after washing the cuvette.

DLS measurements were performed at a scattering angle of 90° using a 662 nm laser. The laser power was automatically adjusted by the instrument to maintain an optimal count rate between $1 \times 10^5$ and $1 \times 10^6$ counts per second. For each sample, 10 acquisitions were recorded with 5 s per acquisition, resulting in a total measurement time of ~1 min per sample.

The autocorrelation functions were analyzed using the DYNAMICS software (version 8.2.0.297, Wyatt Technology) to obtain the size distribution. A regularization algorithm was employed to fit the autocorrelation data, assuming a spherical model for the particles. The intensity-weighted size distribution was converted to a volume-weighted distribution.

**Cryo-electron microscopy (relates to Fig. 1d, and Supplementary Fig. 4).** Freshly prepared suspensions (3 μL) of the identical samples, also subjected to DLS analysis (see above), were pipetted onto glow-discharged, 2/1 holey carbon 200 mesh copper grids (Quantifoil Micro Tools), incubated for 4 sec, and blotted. The grids were then immediately snap-frozen in liquid ethane-propane mixture using a Vitrobot Mark 4 (Thermo Scientific). The grids were then loaded into the autoloader of a 200 kV Glacios cryo-TEM (Thermo Scientific). Grid hole overview images were collected manually using a Falcon4i direct electron detector camera (Thermo Scientific). To increase the signal-to-noise ratio, a median filter (despeckle operation) was applied twice to each overview image before further processing. Ten images from the dox-induced sample were randomly selected for manual vesicle segmentation in Fiji (ImageJ)[30]. Segmented areas were used to estimate the radii of equivalent circular areas. For display, image areas containing the selected vesicles were cropped and contrast-enhanced using the Contrast Limited Adaptive Histogram Equalization (CLAHE) function in Fiji image software (block size 127, histogram bins 256, maximum slope 1.5)[31] and assembled into a montage. High-resolution images of NTVE particles were acquired using a final pixel size of 0.246 nm.

**HiBiT bioluminescence quantification.** For quantification of NTVE particles in culture supernatants, samples were centrifuged at 200 × g for 5 min and then analyzed with the Nano-Glo HiBiT Lytic Detection System (Promega) according to the manufacturer's protocol. Luminescence was measured on a Centro LB 960 plate reader (Berthold Technologies) with 0.5 s acquisition time.

**p24 (HIV-1) quantification (relates to Fig. 1l).** NTVE particles in supernatants were estimated using the commercial Lenti-X p24 Rapid Titer Kit (Takara Bio, 631476). The ELISA kit was used according to the manufacturer's protocol; the reaction with TMB Substrate was stopped after 27 min.

After sterile filtration and concentration, the volume of the supernatants was normalized to 250 μL. Titers were calculated by using the recommended dilution series of recombinant antigens and 1:10 and 1:100 dilutions of the VLP suspension in PBS. To calculate particles per mL, we assumed that 2500 monomers form one particle. Supernatants from cells treated with DMSO instead of dox served as a negative control (Supplementary Fig. 9).

## Bright-field microscopy

Movies of cardiomyocyte contractions were acquired at 60× magnification and 13.8 frames per second on a Zeiss LSM 800 microscope equipped with a standard transmitted-light module. Average image intensity per frame was extracted and analyzed with the find_peaks function in Python (SciPy) to quantify the contraction frequency.

## Fluorescence microscopy

Epifluorescence microscopy images were taken on an EVOS FL Auto Imaging System (Invitrogen, Thermo Fisher Scientific) with identical settings across all samples. Confocal microscopy was conducted on a Leica SP5 system (Leica Microsystems) at 37 °C.

## Flow cytometry

Cells were gently detached in a suitable volume of Accutase, pelleted (200× g, 5 minutes), and resuspended in ice-cold 0.4% formalin for 10 minutes. Fixed cells were pelleted again (200 × g, 5 min) and resuspended in ice-cold 200 μL DPBS. Flow cytometry was performed on the BD FACSaria II system (controlled with the BD FACSDiva Software (v.6.1.3, BD Biosciences). Briefly, the main population of the cells was gated according to their FSC-A and SSC-A. Secondly, single cells were gated using FSC-A and FSC-W. The final gate (green fluorescence) was used to determine the proportion of edited reporter cells after successful mRNA/PE RNP delivery in co-culture.

## Cell fractionation (relates to Fig. 5c)

To separate cytoplasmic and nuclear RNA, cells were fractionated (Fig. 1m, reference). After removal of the cultivation media, cells were rinsed with DPBS (Gibco; Thermo Fisher Scientific) and incubated in Accutase solution (Gibco; Thermo Fisher Scientific) for 10 minutes. The detached cells were collected and centrifuged for 3 minutes at 300× g and 4 °C. Cells were resuspended in 750 μL DPBS (Gibco; Thermo Fisher Scientific) and centrifuged again (300× g at 4 °C). Cell pellets were resuspended in 750 μL ice-cold mild lysis buffer (50 mM Tris, 100 mM NaCl, 5 mM $MgCl_2$ and 0.5% Nonidet P-40, pH 8) and incubated for 5 min to allow lysis of the cell membrane only. The lysate was centrifuged for 2 minutes (18,000 × g at 4 °C), and the supernatant was carefully transferred to a fresh Eppendorf reaction tube. The pelleted nuclei were resuspended in 750 μL mild lysis buffer. Subsequently, 1.5 μL proteinase K (20 mg/mL; Thermo Fisher Scientific) and 4 μL 40% sodium dodecyl sulfate solution were added to both fractions, which were vortexed vigorously before incubation for 15 minutes at 37 °C. Finally, the fractions were centrifuged at 18,000 × g for 2 min to clear impurities from the samples.

## RNA isolation from cell lysates

The Monarch Total RNA Miniprep Kit (NEB) was used to purify RNA from cell lysates. The Quick-RNA 96 Kit (Zymo Research) was used according to protocol for experiments with high sample numbers. RNA was stored at −80 °C.

## RNA isolation from NTVE vesicles

To isolate RNA from NTVE vesicles, supernatants were sterile-filtered to remove the remaining dead cells. Subsequently, the supernatant was concentrated 20x before lysis and purification using the Quick-RNA Viral 96 Kit according to protocol. RNA was stored at −80 °C.

## Affinity purification (relates to Fig. 5 and Supplementary Fig. 3)

40 μL Anti-FLAG® M2 magnetic beads (Merck, M8823) were equilibrated with 200 μL of supplemented DMEM for pull-down of NTVE vesicles obtained from a 6-well plate of HEK293T cells. After one washing step with 400 μL PBS, the sterile-filtered supernatants were added to the beads. Samples were incubated at room temperature with slow rotation for 2 hours. After three washing steps (500 μL PBS), NTVE vesicles were eluted using 100 μL of a 4x FLAG peptide solution (Merck, F3290).

For HA-mediated purification, 35 μL Pierce Anti-HA Magnetic Beads (Thermo Fisher Scientific, 88836) were used. The purification of NTVE vesicles via HA-mediated pull-down was similar to the anti-FLAG purification. Finally, the vesicles were eluted from the beads using 100 μL HA peptide (1 mg/mL). If cells were co-cultivated expressing either dVSV-G(W72A):HA or dVSV-G(W72A):FLAG, the supernatants were first incubated with anti-FLAG beads, followed by anti-HA beads. The anti-FLAG purification flow-through was applied to anti-HA beads.

All elution fractions were analyzed for secretion strength via HiBiT luminescence assay.

## RT-qPCR

RNA samples were subjected to DNAse digestion with DNAse I (NEB) with the following protocol changes: the DNAse concentration was doubled, and the incubation time was extended to 15 minutes, followed by inactivation at 75 °C for 10 min. The Luna Universal One-Step RT-qPCR Kit (NEB) was used per the manufacturer's protocol. Reactions were run in 384-well plates (11 μL per well) in technical duplicates for 45 cycles. Control amplification reactions without reverse transcriptase were performed as a reference for each RT-qPCR run. The reactions were performed and monitored in an Applied Biosystems QuantStudio 12 K Flex Real-Time PCR system. Primer sequences are provided in Supplementary Data 4.

## Illumina library preparation

Library construction was performed as described in the high-throughput protocol of the TruSeq stranded mRNA Sample Prep Guide (Illumina) in an automated manner using the Bravo Automated Liquid Handling Platform (Agilent). Samples were either (poly(A)-enriched via poly(T) beads (Fig. 1f, g, j, k, Fig. 5, and Supplementary Figs. 5, 14, 15, 16), or total RNA, without enrichment, was used in combination with rRNA depletion (Fig. 1e, Supplementary Fig. 5c,d,e).

## Template switching-based library preparation

RNA was either isolated as stated or added directly into the reaction mix for direct reverse transcription (for optional direct lysis, add 10%

**Table 1 | Reaction mix for cDNA generation**

| Component | Volume [μL] |
|---|---|
| 2x SuperFi II PCR master mix | 5.0 |
| mGL-mRNA spike-in (1 pg/μL) | 1.0 |
| RNase inhibitor (optional) | 0.1 |
| TSO (100 μM) 5′- Biosg/AAGCAGTGGTATCAACGCAGAGTACrGrGrG | 0.25 |
| poly(T) primer (100 μM) 5′-/5Biosg/AAGCAGTGGTATCAACGCAGAGTACTTTTTTTTTTTTTTTTTTTTTTTTTTTTTTVN-3' | 0.1 |
| Superscript IV | 0.15 |
| $H_2O$ | 2.4 |
| RNA sample | 1.0 |

**Table 2 | Reaction mix for cDNA tagmentation**

| Component | Volume [µL] |
| --- | --- |
| Loaded Tn5 | 1.2 |
| 5x Transposition buffer mixed with DMF (1:1) | 2.0 |
| PEG 40% | 2.0 |
| H2O | 3.8 |
| cDNA sample | 1.0 |

**Table 3 | Reaction mix for index PCR**

| Component | Volume [µL] |
| --- | --- |
| 2x SuperFi II PCR master mix | 5.0 |
| 7x Pfu or Phusion polymerase | 0.15 |
| H$_2$O | 1.85 |
| Illumina index primer (1 µM) | 2.0 |
| Sample | 1.0 |

Triton X-100). The samples were mixed at a ratio of 1:10 with a reaction master mix containing Superscript IV reverse transcriptase, a poly(T), and a template switching oligo (TSO), Platinum SuperFi II PCR Master Mix (Thermo Fisher Scientific), and 1 pg of mRNA coding from mGreenLantern (mGL) spike-in (Table 1). The samples were incubated for one hour at 50 °C for reverse transcription, followed by standard PCR amplification conditions with either 15 (cell lysates) or 25 (supernatants) cycles. The quantity of the resulting cDNA was assessed using either qPCR or analytical gel electrophoresis with SYBR Gold staining.

The cDNA was tagmented for 8 min at 50 °C using Tn5 transposase loaded with barcoding adapters (Table 2). Tn5 was inactivated with 4 µL 0.2% SDS per sample immediately. The 5x transposition buffer used for the reaction comprised 50 mM Tris (pH 8.2) and 25 mM MgCl$_2$.

For Tn5 loading, P5-i5 universal connectors A and B were annealed separately with a reverse barcoded adapter in 10x annealing buffer, comprised of 100 mM HEPES (pH 7.2), 500 mM NaCl, and 10 mM EDTA. The oligonucleotides were annealed at 95 °C for one minute, followed by a temperature ramp to 25 °C at a rate of 0.1 °C/s. Tn5 was then incubated with the annealed, barcoded adapters for at least 60 min at room temperature. Subsequent amplification of the fragments (Table 3) was primed with 2 µM Illumina index primers (Supplementary Data 4). Sequences of universal P5 i5 adapters are also provided in Supplementary Data 4.

Samples were incubated at 72 °C for 3 min prior to amplification to allow for strand repair, followed by 15 cycles according to standard PCR amplification protocols. Amplicons spanning the range of 200 to 700 bp were purified from agarose gel electrophoresis for Illumina sequencing.

### Quality control of the library

Following reverse transcription and cDNA library construction, an initial quality control step was implemented prior to sequencing. The abundance of several housekeeping/cell-type-specific genes was quantified by qPCR, compared to no−reverse transcription (noRT) controls to confirm RNA-dependent amplification. In parallel or as an alternative, libraries were subjected to nanopore sequencing for early validation, with reads aligned to the human reference transcriptome to verify transcriptomic origin and library complexity before downstream sequencing.

### Illumina sequencing

If not indicated otherwise, samples were either poly(A)-enriched or reverse transcription was primed through a poly(T) oligonucleotide, binding to the 3' of mRNA. The resulting cDNA was PCR amplified after tagmentation, fragments were multiplexed and sequenced as 100 bp paired-end run on an Illumina HiSeq4000 or HiSeq X platform with a depth of ≥30 million reads per sample.

The template switching-based library preparation was applied for Figs. 2–4, 6, 7, and Supplementary Figs. 6, 8–13, 19, 21.

### Quantification of NTVE-exported RNA.

A poly-adenylated transcript encoding mGL was in vitro transcribed using the HiScribe T7 mRNA Kit with CleanCap Reagent AG (NEB, E2080S). The transcript contains a genetically encoded 67 nt poly(A) tail. After quantifying the mRNA amount via Qubit RNA High Sensitivity, 1 pg of mGL transcript was spiked into the sample after the total RNA extraction and before the NGS library preparation. To obtain the absolute RNA amount of a sample, the proportion of fragments mapping to mGL was used for a known amount of mGL spike-in (1 pg).

The amount of exported transcripts per day was calculated from a spike-in experiment three days post-NTVE induction. We assumed that cells divide twice during cultivation and that the average length of an mRNA is -1500 nt[32–34].

### Quantification workflow (v1)

Given the relatively low amount of RNA input from NTVE vesicles, duplicate reads were removed using czid-dedup (v0.1.0) in the paired-end mode to correct for a possible excess of PCR duplicates, followed by several downstream data analysis workflows schematized in the flow chart in Supplementary Fig. 5a. This workflow was used for Figs. 1,3 and Supplementary Figs. 5,14–16.

### STAR-based analysis.

To establish correspondence between NTVE and lysate, we obtained counts at the level of genes and aligned transcripts to a reference (STAR, v2.7.10b_alpha_220111)[35]. RNA-seq paired-end reads were aligned using a combined human-murine index built with assembly of GRCh38 with annotations from Ensembl release 108 for Homo sapiens and GRCm39 with annotations from Ensembl release 109 for *Mus musculus*. To account for reads mapping to the mRNA encoding NTVE$_{PABP}$ and the mGL-encoding mRNA used as a reference of a known amount, the corresponding sequences were added to the index[36]. STAR was run with the --outSAMtype BAM SortedByCoordinate parameter on the paired-end reads. Uniquely mapped fragments were used as inputs to calculate counts with featureCounts (v2.0.3)[37] on the human-murine index, using the following parameters to account for paired-end, reverse-stranded data: "featureCounts -a [combined_index.gtf] -o [output_file] -g gene_id -t exon -s 2 -p". The resulting counts were normalized to counts per million (CPM).

### Salmon-based analysis.

To assess which transcript lengths were exported by NTVE$_{PABP}$, we obtained transcript-level RNA-seq counts by Salmon (v1.10.1)[16]. The index was constructed from *Homo sapiens* cDNA and ncRNA sequences (GRCh38, Ensembl release v108), *Mus musculus* (GRCm39, Ensembl release v109), and *Bos taurus* cDNA and ncRNA sequences (ARS-UCDI1.2, Ensembl release v108), to identify possible contaminations from bovine serum contained in the cell culture media. The sequences for the mRNA encoding NTVE$_{PABP}$ and the mGL-encoding mRNA were added to the index. The resulting counts were normalized to transcripts per million (TPM) to account for differences in sequencing depth across samples. Human transcripts were identified and isolated by filtering for ENST (Ensembl transcript) identifiers. For each gene, we calculated the relative abundance of all expressed isoforms for each sample. We defined a dominant isoform of a gene in a sample as a transcript that accounted for ≥ 90% of the total gene expression (sum of all isoform TPM values) of the gene in the specific sample. For each gene with a dominant isoform, we recorded the transcript identifier, the proportion of total gene expression represented by the dominant isoform, and its normalized abundance in TPM. To plot the mean TPM for bins of transcript

lengths, we stratified transcripts by the length of the corresponding annotation and calculated the mean of the TPM of all transcripts per bin per sample. We then computed the mean and standard deviation across all samples of one condition. For analyses where only coding genes were selected, we renormalized the TPMs obtained from Salmon again to 1 million after subselection.

**Positional read coverage analysis.** We filtered for dominant isoforms as outlined above to then compute the read coverage with a pipeline implemented in Python (v3.12.3) utilizing the pandas data analysis library for efficient data manipulation and processing. For the calculation of the coverage depth, gene annotations were obtained from the Ensembl annotation GTF file for *Homo sapiens* (v108) containing genomic coordinates and feature information[38]. The pipeline filtered the GTF annotations to retain only exon features corresponding to the identified dominant isoforms.

We extracted and compiled the genomic coordinates, including chromosome, start position, end position, and strand information. The coordinates were processed to create a BED format file containing all exonic regions of the dominant transcripts.

Samtools (v1.19.2) was used with the depth command to calculate the read depth for the corresponding exonic features using "samtools depth -b dominant_isoform_genes_coordinates.bed -@ 8 -o [output_file] [input_bam]". Coverage depths were normalized to account for differences in sequencing depth and transcript abundance. The normalization process involved scaling each transcript's coverage by its total read count to obtain position-wise proportions, multiplying by the transcript length to maintain relative depth information. For visualization, transcripts were aligned by the 3′ end of the annotation, and coverage was analyzed across the last 5,000 nt.

**Assessment of binary classifier performance.** We evaluated the performance of NTVE$_{PABP}$ as a binary classifier by establishing ground truth labels from lysate samples and assessing prediction accuracy using supernatant samples from the same cells 96 hours after dox induction and 72 hours after IFNγ stimulation (or vehicle control). Ground truth labels were determined using the DESeq2 hypothesis testing framework on lysate samples, comparing six replicates without stimulation and three with IFN-γ stimulation. Genes were classified into two sets: a positive set comprising genes with significant absolute log2 fold changes greater than 1 (FDR < 0.05, using the 'greaterAbs' alternative hypothesis option) and a negative set containing genes with significant absolute log2 fold changes less than 1 (FDR < 0.05, with the 'lessAbs' alternative hypothesis option). This classification yielded 39 positive and 16,095 negative gene IDs. Log2 fold changes were shrunk using the 'ashr' method to mitigate effect size overestimation in low-count genes, and p-values were adjusted for multiple testing using the Benjamini-Hochberg procedure. For prediction, we analyzed matched supernatant samples using DESeq2's greaterAbs test to identify genes with absolute log2 fold changes greater than 1 between the same treatment conditions (six dox replicates *vs.* three IFNγ replicates). The adjusted p-values from this analysis served as prediction scores. Classifier performance was evaluated using Receiver Operating Characteristic (ROC) and Precision-Recall (PR) curves, with corresponding Area Under the Curve (AUC) values calculated using the sklearn.-metrics package in Python.

**Differential expression analysis.** Differential expression estimation was performed using the DESeq2 package (v1.34.0)[39] in R (v4.1.2). P-values were adjusted for multiple testing using the Benjamini-Hochberg procedure to control the false discovery rate (FDR). Log2 fold changes were shrunk using the 'apeglm' method[40] to reduce the effect size estimates of low-count genes. Analyses were performed separately for lysate and supernatant samples, comparing dox-treated samples against DMSO (non-induced), CRISPRa *ASCL1* transactivated,

and IFNγ-stimulated samples. For the IFNγ-stimulated condition, six dox-treated replicates were compared against three replicates after IFNγ stimulation for both lysate and NTVE. For CRISPRa *ASCL1* transactivated samples, the DE calculation was conducted in triplicate versus reference triplicates for both NTVE and lysate. The effects of DMSO were assessed in the lysate.

**Filtered quantification workflow (v2)**
To account for the lower RNA input of the experiments shown in Figs. 2, 4, 6, and corresponding Supplementary Figs. 6, 7, 8, 10, 11, 12, 13, 19, 20, 21, we optimized our processing pipeline based on the alignment software minimap2 (v2.30)[41] and the transcript quantification software Salmon (v1.10.0)[16], together with a custom filtering program to avoid repetitive reads that could appear as artefacts of amplification (Supplementary Fig. 10a).

Short read alignment was performed against the reference transcriptome derived from the Ensembl releases 108 and 109, generated by merging the cDNA and ncRNA, as well as our synthetic sequences (HIV-*gag*, mGL for the spike ins)

Minimap2 was used in the default short read configuration (minimap2 -ax sr -t 8 <reference.fa.gz> <R1.fastq.gz> <R2.fastq.gz> | samtools view -bS -f 0×2 - > output.bam), with SAMtools (v1.16.1)[42] retaining only the properly mapped reads. After alignment, SAMtools is invoked a second time to sort the alignments by name using samtools sort -n -@ 8 -o <output_namesorted.bam> <input.bam>.

Following the sorting step, a k-mer-based filter is employed to eliminate low-quality reads (see Supplementary Fig. 10b). This filter operates by first extracting all k-mers from each read and tabulating them to create a k-mer dictionary. It then calculates the proportion of uniquely occurring k-mers. Read pairs are preserved if this proportion exceeds a predetermined threshold. Furthermore, the filter can optionally examine the Compact Idiosyncratic Gapped Alignment Report (CIGAR string) to verify the length of the longest consecutive mapped stretch, with our standard setting requiring at least 80 consecutive mapped base pairs.

Quantification of filtered alignments is conducted using Salmon, which utilizes the identical concatenated reference transcriptome as minimap2. This quantification is performed at a per-transcript level, adjusting for biases related to GC content and transcript position. For every sample, a table is generated where each row represents a transcript, uniquely identified by its transcript identifier. This table includes key metrics such as read count and TPM (Transcripts Per Million). Supplementary data, such as a Boolean flag for protein-coding status, an associated gene identifier, and a gene symbol, can be derived from the corresponding GTF file.

**Determination of export ratio.** Protein-coding gene-level expression information was grouped as matched supernatant-lysate pairs. Genes with lysate aggregated CPM/TPM < 1 were excluded. Export ratios were calculated as supernatant divided by lysate expression level. Export ratios were log10-transformed for visualization. The distribution is shown in histograms with 70 bins. For each condition, the mean density across biological replicates was plotted with 95% confidence intervals.

**mRNA detection rate across expression quantiles.** We performed a quantile-based rediscovery analysis of genes detected in the lysate and in the supernatant (Fig. 2). Gene-level aggregated TPM values of endogenous protein-coding genes (excluding mitochondrial genes) were ranked after excluding genes with aggregated TPM < 1. The expression levels were ranked by expression in the source compartment and divided into 50 evenly spaced quantiles. The detection rate was calculated as the percentage of genes within that quantile that were detected with TPM > 0 in all three replicates of the target compartment.

**Supernatant–lysate correlation by transcript length.** To assess the fidelity of RNA export variation with transcript length, we analyzed the Spearman correlation between supernatant and lysate transcript abundance divided into length bins (Fig. 2). Analysis was performed on protein-coding transcripts with TPM > 1 in all replicates ($n = 3$) of both the supernatant and the lysate fraction. For each length bin and sample SN-Lysate pair, transcripts were binned by length, and the Spearman correlation coefficient was calculated between all pairwise combinations of supernatant and lysate. Mean and standard deviation across replication pairs were calculated. Mitochondrial transcripts were excluded from the analysis.

**Coverage analysis of highly abundant transcripts.** To analyze the spatial distribution of reads within transcripts, we select the 5000 most abundant transcripts based on the read counts as determined by the 2-step quantification pipeline of a representative sample.

We used samtools depth -a to calculate the per-position coverage depth for all transcripts across the analyzed samples.

The coverage for each transcript j in sample k was normalized by Area-Under-the-Curve (AUC) normalization. The normalized coverage C' at position i is defined as

$$C'_{ijk} = \frac{C_{ijk}}{\sum_m^{L_j} C_{mjk}} \quad (1)$$

with C the raw per-position coverage as derived from samtools depth and $L_j$ the total exonic length of the transcript j. This normalization is performed up to 3000 base pairs upstream of their transcription end site (TES). The transcripts were at their 3' end, and the normalization was rescaled to correspond to the transcript length, i.e., the normalized coverage per position is on average 1.

The aligned coverages were aggregated into discrete spatial bins of 10 base pairs extending to 3000 base pairs upstream of the TES. For each bin, the mean normalized coverage was calculated by averaging across all analyzed transcripts and replicates within a sample group.

**Coverage ratios.** We calculate the relative apparent enrichment by dividing the mean binned coverage of the supernatant and lysate-derived fractions. This gives us the relative enrichment or depletion at specific distances from the TES.

**UMAP.** The UMAP embedding was performed using scanpy (1.11.1) on measurements with over 100k aligned reads. The initial feature vector was the expression of all 20,036 protein-coding genes. Samples with a total read count below $10^5$ reads were excluded from analysis. This high-dimensional data was first reduced to 41 features using principal component analysis (PCA). Subsequently, a k-nearest neighbor graph ($k = 15$) was constructed in this lower-dimensional PCA space, which was then used for the final UMAP embedding.

**Gene set enrichment analysis (GSEA).** We implemented a Dockerized version of fgsea (v1.32.4) for performing gene set enrichment analysis. We downloaded the Hallmark and C5 data set from the GSEA MSigDB, corresponding to the hallmark pathways and the gene sets derived from the gene ontology resource[43–45].

**Discovery of genes with lineage-specific expression patterns**
To identify genes with lineage-specific expression patterns, we used DESeq2. We were interested in patterns that distinguish one of the three lineages (endoderm, mesoderm, ectoderm) from the remaining two (test-vs-pool). For each target lineage, we modeled the gene expression of that specific lineage, comparing it to a model fitted to the remaining two pooled lineages combined using a likelihood-ratio test.

**One-vs-rest likelihood-ratio test (relates to Fig. 6).** We model raw counts using a negative binomial generalized linear model with a log link. For two nested models $F$ and $R$. For each gene, we compare a full model $F$ and a reduced model $R$ using a likelihood-ratio test,

$$\Lambda = 2\,(\log L(F) - \log L(R)) \quad (2)$$

where $L(F)$ denotes the maximized negative binomial likelihood under the full model and $L(R)$ the reduced model. The test statistic $\Lambda$ is evaluated against a $\chi^2$ distribution with degrees of freedom equal to the difference in the number of free parameters between $F$ and $R$, yielding a one-vs-rest test for lineage-specific expression. Let $K_{jg}$ denote the observed count for a given gene g in sample j, and let $\mu_{jg}$ be its expected value. The full sample identity consists of lineage, sampling time point, and replicate number. The hypotheses tested for each gene were:

$H_0$: There is no difference in expression between the tested and pooled lineages.

$H_1$: Expression differs between the tested lineage and the pooled lineages.

We defined a binary indicator variable $L_j \in \{0, 1\}$, where $L_j = 1$ if sample j belongs to the tested lineage and $L_j = 0$ otherwise. The two nested models were defined as:

$$F : \log \mu_j = \log s_j + \beta_0 + \beta_1 L_j \quad (3)$$

$$R : \log \mu_j = \log s_j + \beta_0 \quad (4)$$

$K_{jg} \sim NB(\mu_{jg}, \alpha_g)$. Here, $s_j$ denotes a sample-specific size factor accounting for library depth, $\beta_0$ is the baseline expression level shared across all lineages, and $\beta_1$ represents the lineage-specific expression effect for the tested lineage.

**Lineage-specific spline model.** To allow for genes with significantly lineage-dependent temporal expression patterns, we used a natural spline model to describe the time-dependent gene expression pattern of the different lineages. Let t denote the time point of lineage j, and let $b_k(t)$ be a natural cubic spline basis of order K evaluated at t. The implementation of this in DeSeq2 is based on example code from ImpulseDE2[46].

To allow for lineage-specific temporal expression patterns, we extended the shared spline model of Fig. 6 by including interaction terms between lineage membership and the spline basis functions. The full model allowing lineage-specific temporal trajectories was defined as:

$$\begin{aligned} F : \log \mu_j(t) = {} & \log s_j + \beta_0 + \beta_1 L_j \\ & + \sum_{k=1}^{K} \gamma_k b_k(t) + \sum_{k=1}^{K} \delta_k L_j b_k(t) \end{aligned} \quad (5)$$

where $\beta_0$ is the baseline expression level, $\beta_1$ represents a constant expression difference between the tested lineage and the pooled lineages, $\gamma_k$ are spline coefficients describing the shared temporal expression pattern, and $\delta_k$ are lineage-specific spline coefficients capturing deviations from the shared trajectory.

The corresponding reduced model enforcing a shared temporal pattern across lineages was:

$$R : \log \mu_j(t) = \log s_j + \beta_0 + \beta_1 L_j + \sum_{k=1}^{K} \gamma_k b_k(t) \quad (6)$$

The null hypothesis tested was:

$$H_0 : \delta_1 = \delta_2 = \cdots = \delta_K = 0 \quad (7)$$

corresponding to identical temporal expression patterns between the tested lineage and the pooled lineages.

**Longitudinal transcriptome analysis of cardiomyocyte differentiation (relates to Fig. 7)**

We used a natural spline model, with the full model allowing specific temporal trajectories defined as:

$$F : \log \mu_j(t) = \log s_j + \beta_0 + \sum_{k=1}^{K} \gamma_k b_k(t) \tag{8}$$

where $\beta_0$ represents a replicate expression factor, and $\gamma_k$ are spline coefficients describing the temporal expression pattern. The corresponding reduced model of constant expression was:

$$R : \log \mu_j(t) = \log s_j + \beta_0 \tag{9}$$

**Statistics & Reproducibility**

Statistical analyses were performed in Python or R (DESeq2, fgsea) and are fully reproducible from the deposited code; detailed workflows are described in the respective quantification sections. All experiments were conducted with at least three biologically independent replicates ($n = 3$) unless otherwise stated in the figure legends, where reduced sample sizes ($n = 1$ or $n = 2$) are explicitly noted. Statistical analyses were performed in Python or R (DESeq2, fgsea) and are fully reproducible from the deposited code; detailed workflows are described in the respective quantification sections.

The experiments were not randomized, and the investigators were not blinded to allocation during experiments and outcome assessment. All experiments were conducted with at least three biologically independent replicates unless otherwise stated in the figure legends, where reduced sample sizes are explicitly noted. We excluded data points from RT-qPCR analysis in Supplementary Fig. 18 due to unspecific amplification, which was detected via melting curve analysis.

**Reporting summary**

Further information on research design is available in the Nature Portfolio Reporting Summary linked to this article.

## Data availability

The raw RNA sequencing data generated in this study have been deposited in the Gene Expression Omnibus (GEO) database under accession code GSE283136. The source data generated in this study are provided in the **Source Data** file. The genetic constructs are available *via* Addgene (#254460-254465). The processed data generated in this study have been deposited in the Zenodo database under accession code 19370054[47]. The interactive plot of the tri-lineage differentiation of Fig. 6 is available at https://magro.codeberg.page/Trilinplots/. Source data are provided with this paper.

## Code availability

Scripts for the processing workflows are available at https://github.com/ggwlab/NTVE and were schematized in Supplementary Figs. 5a and 10a. The code is also available at Zenodo under accession code 19370054[47].

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

## Acknowledgements

We gratefully acknowledge the financial support provided by the European Innovation Council (EIC) Pathfinder project under grant number 101115574 (N.A., J.Ge., G.G.W), and the ERC PoC under grant number 101138939 (G.G.W.). We thank Dr. Inti de la Rosa Velázques and Dr. Gertrud Eckstein of the Core Facility Genomics of Helmholtz Munich for library preparation and NGS. We thank Dr. Carsten Peters of the TUM Electron Microscopy Facility for the help with cryo-EM data acquisition. We also thank the Groll Laboratory at TUM for access to their DLS instrument. This work was supported by the Impulse and Networking Fund of the Helmholtz Association (Helmholtz Excellence Network BioEM, grant EXNET-01-14).

## Author contributions

N.A. developed methodology, designed genetic constructs, designed and conducted the cell and biochemical experiments, generated stable cell lines, generated schematics and figures, and co-wrote the manuscript; M.G. analyzed and visualized the NGS data, co-designed the perturbation experiments, and conducted the IFN-gamma stimulation experiments, established the AAV system and the low-input sequencing workflows, and conducted the experiments in primary neurons; J.Ge developed methodology, designed genetic constructs, co-conducted experiments on barcoded RNA export, generated stable cell lines, designed and conducted the cell-cell-communication experiments, and drafted some of the respective accompanying figures; L.S. worked with N.A. on the biochemical experiments; M.J. worked on generating hiPSC lines expressing NTVE and co-analyzed the cardiac differentiation data; H.L. worked on the comparison of NTVE to other methods for RNA export, on the trilineage differentiation, and the cardiac differentiation; T.P. worked with N.A. on the RT-qPCR analysis and the ASCL1 transactivation; J.W. optimized the library preparation with M.G.; L.G. established the AAV system and conducted the experiments in primary neurons with M.G.; S.Schm. worked with N.A. on the differentiation of hiPSC; T.O. and E.R performed the trilineage differentiation and end-point characterization of the SCTi003-A line; E.S., F.W., N.W., M.S., A.S., and T.S. assisted with cell and biochemical experiments; A.G., optimized components for the library preparation, O.B. obtained and analyzed the cryo-EM data; S.B. assisted with expression in primary neurons; A. M. advised on the use of the MRIi003-A line; St.Schn. helped with statistical analysis; F.T. advised population-specific experiments and data analysis; J.Ga. advised on perturbation experiments and methodology, reviewed the text and NGS analyses; D-J.J.T developed methodology, designed genetic constructs, designed experiments, supervised the research project, reviewed the data analysis and presentation; G.G.W. conceptualized, initiated, and secured funding for the research program, developed methodology, designed experiments, reviewed data analysis and presentation, supervised and administered research, and wrote the manuscript with input from co-authors.

## Funding

## Competing interests

The authors declare no competing interests.
