## [Transparent Peer Review file · Nature Communications]

Non-destructive transcriptomics via vesicular export

Corresponding Author: Professor Gil Westmeyer

Version 1:

Reviewer comments:

Reviewer #1

(Remarks to the Author)

The authors have added a lot more data and experiments, which has improved the manuscript substantially. Only minor comments remain:

In Figure S1 – a “-” sign is missing for the left columns (only “+” is shown for the right ones)

In Figure 1E – its not clear enough the the color of “nuclear” corresponds to the violin plot whereas the mitochondrial genes are shown as dots – the legend should be clearer.

Figure S12 – the gray line should be explained in the legend

Figure 3C – is the gray random classifier line at $y=0$ in the precision-recall plot? This appears strange: it should indeed be horizontal, but it should presumably be at $y>0$ since even a random classifier should be correct at some rate.

(Remarks on code availability)

Reviewer #2

(Remarks to the Author)

In this revised manuscript the authors provide the requested data, especially the amounts of RNA released, and the amounts of transcripts per released particle, as well as additional control experiments. Overall, the strong and interesting manuscript is now further improved and the main questions from the earlier round of review are addressed. I have no additional comments or questions.

(Remarks on code availability)

REVIEWERS' COMMENTS

Reviewer #1 (Remarks to the Author):

The authors have added a lot more data and experiments, which has improved the manuscript substantially. Only minor comments remain:

In Figure S1 – a “-“ sign is missing for the left columns (only “+” is shown for the right ones)

- Thank you, we fixed this mistake.

In Figure 1E – its not clear enough the the color of “nuclear” corresponds to the violin plot whereas the mitochondrial genes are shown as dots – the legend should be clearer.

- We optimized the figure legend.

Figure S12 – the gray line should be explained in the legend

- We included this description.

Figure 3C – is the gray random classifier line at $y=0$ in the precision-recall plot? This appears strange: it should indeed be horizontal, but it should presumably be at $y>0$ since even a random classifier should be correct at some rate.

-Thank you for this observation. The precision is 5%. We have clarified this in the figure legend.

Reviewer #2 (Remarks to the Author):

In this revised manuscript the authors provide the requested data, especially the amounts of RNA released, and the amounts of transcripts per released particle, as well as additional control experiments. Overall, the strong and interesting manuscript is now further improved and the main questions from the earlier round of review are addressed. I have no additional comments or questions.

- We thank the Reviewers again for all of their helpful comments and suggestions.